# REVIS: Sparse Latent Steering to Mitigate Object Hallucination in Large Vision-Language Models

**Jialin Wu** [1]  **Wei Shi** [1]  **Han Shen** [1]  **Peigui Qi** [1]  **Kunsheng Tang** [1]  **Zhicong Huang** [1]  **Binghao Wang** [1]  **Zhou Yang** [1]

## Abstract

Despite the advanced capabilities of Large Vision-Language Models (LVLMs), they frequently suffer from object hallucination. One reason is that visual features and pretrained textual representations often become intertwined in the deeper network layers. To address this, we propose REVIS, a **training-free framework** designed to explicitly re-activate this suppressed visual information. Rooted in latent space geometry, REVIS extracts the pure visual information vector via orthogonal projection and employs a calibrated strategy to perform **sparse intervention** only at the precise depth where suppression occurs. This surgical approach effectively restores visual information with **minimal computational cost**. Empirical evaluations on standard benchmarks demonstrate that REVIS reduces object hallucination rates by **approximately 19%** compared to state-of-the-art baselines, while preserving general reasoning capabilities.

## 1. Introduction

Despite the remarkable capabilities of Large Vision-Language Models (LVLMs), they suffer from a persistent reliability bottleneck: object hallucination. This phenomenon, where models generate plausible but non-existent visual details, fundamentally stems from a conflict between external visual information and internal parametric knowledge (Liu et al., 2024b;d). When visual information is weak or ambiguous, the model's decoding trajectory drifts from grounded perception to ungrounded probabilistic guessing, over-relying on the language priors inherent in the LLM backbone.

Current mitigation strategies predominantly operate as exter-

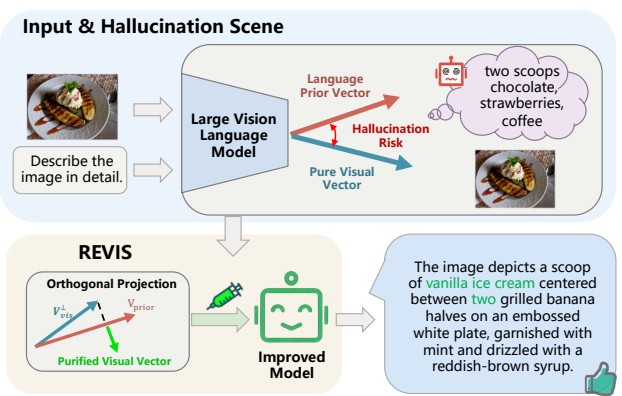

Figure 1. Overview of REVIS. After applying orthogonal projection to isolate the pure visual vector from language priors, REVIS steers the LVLM to generate faithful and grounded descriptions, effectively correcting the initial hallucinations (*e.g.*, "chocolate", "strawberries").

nal patches rather than internal cures. Training-time alignment methods, such as LLaVA-RLHF (Sun et al., 2024) or HA-DPO (Zhao et al., 2023), rely on constructing high-quality preference datasets and require computationally expensive retraining, limiting their flexibility. Conversely, inference-time interventions, like Contrastive Decoding (CD) (Leng et al., 2024), attempt to suppress hallucinations by manipulating final output logits. While effective at surface-level correction, these methods incur significant inference latency, often requiring multiple forward passes or auxiliary models. Furthermore, by operating solely on the final probability distribution, they act as post-hoc filters rather than resolving the underlying representational conflict within the model's processing layers.

In this work, we diverge from surface-level suppression to sparse latent steering, as illustrated in Figure 1. We hypothesize that, particularly in LVLMs adapted from pre-trained LLMs, the root cause of hallucination involves the active suppression of visual information by dominant **text inertia** (Liu et al., 2024d) or **language prior** (Favero et al., 2024), which represent the model's intrinsic propensity to revert to **ungrounded probabilistic guessing** when visual information is overshadowed. Our preliminary analysis

[1]Ant Group. Correspondence to: Jialin Wu <jinlin.wjl@antgroup.com>, Han Shen <sh480096@antgroup.com>.

*Proceedings of the 43rd International Conference on Machine Learning*, Seoul, South Korea. PMLR 306, 2026. Copyright 2026 by the author(s).

(Section 3) supports this but highlights a critical challenge: while factual and hallucinatory states are linearly separable in deep layers, the raw difference vector remains entangled with these language priors. Consequently, naively amplifying this direction boosts both visual information and the entangled language priors, often driving the model into degenerate states (model collapse). However, we demonstrate through causal probing (Section 3.2) that *purified* latent directions exhibit superior stability. Unlike entangled vectors that cause collapse, our purified vector allows for high-intensity intervention without degradation, confirming that a robust causal link between latent directions and visual information exists provided the entanglement is resolved.

Guided by these insights, we introduce REVIS[1] (**RE**-activating **VIS**ual information), a training-free sparse latent steering framework designed to surgically restore visual awareness. Unlike previous steering methods that operate on entangled representations (Liu et al., 2024c), REVIS employs an orthogonal visual steering mechanism rooted in the geometry of the latent space. We decouple the visual information from **hallucination-inducing language prior** via orthogonal projection, extracting a pure visual information vector. This ensures that the intervention enhances visual grounding without unintentionally amplifying the **language prior** that leads to hallucination. Furthermore, recognizing that the separability of factual and hallucinatory states is non-uniform across layers, we employ a **sparse intervention** strategy. By leveraging calibration data, we automatically identify the optimal layer where visual information exerts the most discriminative influence.

At inference time, REVIS dynamically monitors the latent "hallucination risk" and activates only when the model drifts away from the visual information subspace, achieving effective mitigation with **minimal computational cost**.

The main contributions of this work are summarized as follows:

- We identify feature entanglement in the latent space as a primary mechanistic cause of hallucination and validate, through causal probing, that pure visual information vectors can explicitly re-activate visual awareness.

- We introduce REVIS, a training-free framework that mathematically decouples visual information from language priors via orthogonal projection and employs sparse intervention to precisely intervene at the optimal depth.

- Extensive experiments on five standard benchmarks and seven models demonstrate that our method achieves superior performance and efficiency compared to existing baselines.

[1] https://github.com/antgroup/Revis

## 2. Related Work

### 2.1. Hallucination Mitigation in LVLMs

Current strategies are primarily categorized into training time alignment and inference time intervention.

**Training Time Alignment.** Methods like LLaVA-RLHF (Sun et al., 2024) and HA-DPO (Zhao et al., 2023) optimize parameters via RLHF or DPO to penalize non-factual tokens. Recently, OPA-DPO (Yang et al., 2025c) addresses off-policy shifts by fine-tuning on expert corrections. While effective, these methods incur significant training costs and rely heavily on large-scale, high-quality preference datasets, limiting their usage in resource-constrained scenarios.

**Inference Time Intervention.** Contrastive Decoding approaches (*e.g.*, VCD (Leng et al., 2024), M3ID (Favero et al., 2024)) suppress language priors by contrasting predictions against negative constraints. However, they fundamentally **double the inference latency** by requiring multiple forward passes. Alternatively, Logit Calibration strategies (*e.g.*, AGLA (An et al., 2025), ONLY (Wan et al., 2025)) adjust output probabilities via heuristics. These methods often depend on fragile hyperparameters or auxiliary detectors, limiting robustness. Unlike these overhead-heavy approaches, REVIS offers a lightweight solution.

REVIS diverges by directly steering internal activations via sparse intervention rather than manipulating final logits. This parsimonious design addresses the root cause of hallucinations with minimal overhead, avoiding the latency penalty of parallel decoding.

### 2.2. Mechanistic Interpretability

Mechanistic interpretability aims to reverse-engineer model components (Meng et al., 2022). Building on this, Activation Steering (Zou et al., 2023; Li et al., 2023a) controls generation via hidden state manipulation. Recently, Lindsey et al. (Lindsey, 2026) expanded this to *causal validation*, using "Concept Injection" to prove that steering vectors can explicitly trigger internal states, linking latent geometry to physical semantics.

In the VLM domain, VTI (Liu et al., 2024c) applies steering to enhance visual stability. However, it operates on *entangled* representations and relies on computationally intensive perturbation averaging. Crucially, VTI lacks a geometric mechanism to precisely isolate visual information from language priors.

REVIS advances this by integrating causal validation with an orthogonality-based framework. By mathematically decoupling visual and textual information, we enable precise "neurosurgery" to re-activate visual awareness in real-time, eliminating the need for multiple forward passes.

*Table 1.* **Definition of the 5-Cluster Semantic State Space.** We construct five distinct counterfactual scenarios to isolate the mechanistic roots of hallucination. $\mathbf{x}$ denotes the visual input, $\emptyset$ denotes the absence of visual input, and Q is the query. **Note on Text Sources:** The *Ground-truth* and *Hallucinated* captions are sourced from the Nullu dataset (Yang et al., 2025b). The *Refusal* response is sampled from a randomized template pool to prevent overfitting (see details in Appendix C.2). All states are extracted via **force-decoding**.

| State Notation | Input Condition | | Caption Source | Description |
| --- | --- | --- | --- | --- |
| | **Image** | **Text** | | |
| $\mathbf{h}_{\text{gt}}$ | $\mathbf{x}$ | Q | Dataset (GT) | Visual input $\mathbf{x}$ + Force-decoded ground-truth caption. |
| $\mathbf{h}_{\text{hall}}$ | $\mathbf{x}$ | Q | Dataset (Hall) | Visual input $\mathbf{x}$ + Force-decoded hallucinated caption. |
| $\mathbf{h}_{\emptyset\_\text{gt}}$ | $\emptyset$ | Q | Dataset (GT) | No visual input $\emptyset$ + Force-decoded ground-truth caption. |
| $\mathbf{h}_{\emptyset\_\text{hall}}$ | $\emptyset$ | Q | Dataset (Hall) | No visual input $\emptyset$ + Force-decoded hallucinated caption. |
| $\mathbf{h}_{\emptyset\_\text{unk}}$ | $\emptyset$ | Q | Refusal Pool | No visual input $\emptyset$ + Force-decoded refusal response. |

## 3. Preliminary Analysis

We ground the theoretical causes of hallucination, specifically visual information deficiency and language prior reliance (Liu et al., 2024b; Li et al., 2023b), in the latent space geometry of LVLMs. In this section, we analyze the separability of factual states, demonstrate the feature entanglement in naive steering vectors, and validate the causal efficacy of our orthogonalized approach.

### 3.1. Feasibility Validation

To investigate whether the model internally distinguishes between factual reasoning and hallucination, we conduct an analysis using a dataset derived from Nullu (Yang et al., 2025b). Specifically, we randomly sample 1,000 instances, where each sample consists of an image, a ground-truth caption, and a hallucinated caption. Based on these samples, we employ a **Counterfactual Input Construction** protocol: for each image-text pair, we extract the hidden states of the last token (*i.e.*, [EOS]) under five distinct conditions (Table 1). Detailed construction methods are provided in Appendix C.1. We select the [EOS] token to capture the global semantic state, ensuring that observed separations stem from intrinsic truthfulness rather than token-level lexical variations.

**Observation.** We visualize the evolution of these hidden states across network layers using t-SNE. Taking Qwen2.5-VL as a representative example, we observe a distinct geometric progression: while shallow layers exhibit significant overlap between factual and hallucinatory states, a clear separation emerges in the deep layers (*e.g.*, Layer 27 shown in Figure 2). The complete layer-wise visualization is provided in Figure 6 in Appendix D.

**Implication.** This emergent separability in deep layers confirms the feasibility of latent steering. It indicates that the model holds distinct representations for visual-grounded generation versus hallucinatory generation. Consequently, hallucination can be modeled as a state selection failure, by a drift from the factual state ($\mathbf{h}_{\text{gt}}$) to the hallucination state ($\mathbf{h}_{\text{hall}}$). This insight serves as the foundation for our method, where we introduce a steering vector to counteract this drift.

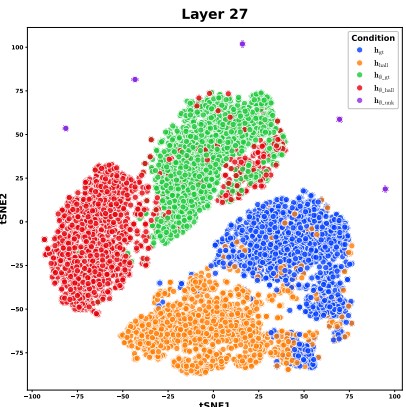

*Figure 2.* t-SNE visualization of **[EOS] token** hidden states at Layer 27 of Qwen2.5-VL. The [EOS] token captures global semantics, verifying that state separation is driven by content factuality rather than lexical variations.

### 3.2. Analysis of Feature Entanglement

Given the separability observed in Section 3.1, latent steering presents a viable mitigation strategy. Existing methods like VTI (Liu et al., 2024c) employ a dual-stage intervention. Specifically, they pre-compute a global steering vector $\mathbf{v}_{\text{text}}$ by averaging the difference between factual and hallucinatory representations on a held-out reference set $\mathcal{D}_{\text{ref}}$. During inference, these **static** vectors are injected into the hidden states $\mathbf{h}^{(\ell)}$ across **all layers**:

$$\tilde{\mathbf{h}}^{(\ell)} = \mathbf{h}^{(\ell)} + \alpha \cdot \mathbf{v}_{\text{text}}^{(\ell)}, \quad \forall \ell \in \{1, \ldots, L\}$$

where $\mathbf{v}_{\text{text}}^{(\ell)} = \mathbb{E}_{\mathcal{D}_{\text{ref}}}[\mathbf{h}_{\text{gt}}^{(\ell)} - \mathbf{h}_{\text{hall}}^{(\ell)}]$. We implemented this framework to evaluate its impact. As shown in Table 2, while VTI effectively reduces hallucinations (CHAIR$_S$: $14.00\% \to 9.00\%$), it incurs a severe penalty on general reasoning (MM-Vet: $70.18 \to 56.38$).

This significant degradation suggests that steering with entangled vectors interferes with the model's core reasoning capabilities. Motivated by the finding that hallucination stems primarily from visual deficiency (Liu et al., 2024b), we propose shifting from error suppression to *visual reactivation*. We define a raw visual vector, $\mathbf{v}_{\text{raw}}^{(\ell)}$, capturing

*Table 2.* **The Trade-off of Latent Steering (VTI).** VTI reduces hallucinations but degrades general reasoning (MM-Vet), CHAIR result with *max token = 64*.

| Method | CHAIR$_S$ ($\downarrow$) | CHAIR$_I$ ($\downarrow$) | MM-Vet ($\uparrow$) |
|---|---|---|---|
| Regular | 14.00 | 6.42 | 70.18 |
| VTI | 9.00 | 3.91 | 56.38 |

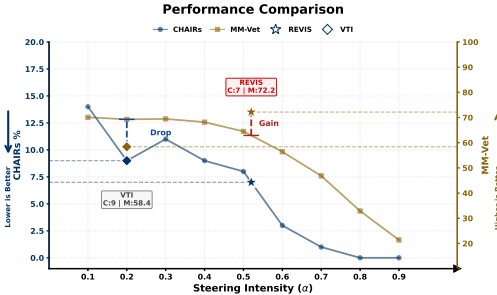

*Figure 3.* Sensitivity analysis of $\alpha$. Naive steering leads to model collapse (sharp metric drop) at high intensities.

the net visual contribution by subtracting the blind state from the standard state at layer $\ell$:

$$\mathbf{v}_{\text{raw}}^{(\ell)} = \mathbb{E}_{\mathcal{D}_{\text{ref}}}[\mathbf{h}_{\text{gt}}^{(\ell)} - \mathbf{h}_{\emptyset\_\text{gt}}^{(\ell)}] \tag{1}$$

Injecting $\mathbf{v}_{\text{raw}}^{(\ell)}$ aims to reinforce visual grounding. While it outperforms VTI at lower intensities ($\alpha < 0.4$), demonstrating the validity of the visual direction, empirical sensitivity analysis (Figure 3) reveals critical instability. Performance deteriorates rapidly as the steering intensity $\alpha$ exceeds 0.5. At $\alpha \approx 0.7$, the model undergoes **collapse**: although CHAIR metrics technically drop to zero, this results from degeneration (*e.g.*, infinite repetition, empty outputs) rather than improved factuality.

We attribute this collapse to Feature Entanglement. We analyze the geometric relationship between $\mathbf{v}_{\text{raw}}^{(\ell)}$ and the **language prior**. Consistent with our definition in Section 1, we operationalize this prior as:

$$\mathbf{v}_{\text{prior}}^{(\ell)} = \mathbb{E}_{\mathcal{D}_{\text{ref}}}[\mathbf{h}_{\emptyset\_\text{hall}}^{(\ell)} - \mathbf{h}_{\emptyset\_\text{unk}}^{(\ell)}] \tag{2}$$

Here, $\mathbf{h}_{\emptyset\_\text{hall}}$ represents the state where the model resorts to probabilistic guessing (hallucinating) without visual input, while $\mathbf{h}_{\emptyset\_\text{unk}}$ represents a conservative "I don't know" response. The difference isolates the specific **direction of ungrounded guessing**, separating it from valid linguistic capabilities. As visualized in Figure 6 in Appendix D, we observe high cosine similarity between these vectors $\mathbf{v}_{\text{raw}}^{(\ell)}$ and $\mathbf{v}_{\text{prior}}^{(\ell)}$ in deep layers. This confirms that $\mathbf{v}_{\text{raw}}^{(\ell)}$ is not purely visual; it remains entangled with language priors. Consequently, increasing $\alpha$ amplifies these priors, leading to the model collapse observed in Figure 3. In contrast, our orthogonalized vector ($\mathbf{v}_{\text{vis}}^{\perp(\ell)} = \mathbf{v}_{\text{raw}}^{(\ell)} \perp \mathbf{v}_{\text{prior}}^{(\ell)}$) effectively reduces CHAIRs while preserving MM-Vet performance.

**Key Insight: The Necessity of Orthogonality.** Naive steering vectors remain entangled with language priors, leading to collapse. To re-activate visual information without corrupting general reasoning, it is necessary to decouple the visual information via orthogonalization, which will be introduced in Section 4.

**Validation of Vector Efficacy and Stability.** To validate the quality of the orthogonalized vector $\mathbf{v}_{\text{vis}}^{\perp(\ell)}$, we assess both its quantitative impact and behavioral stability. As evidenced in Table 3 and Table 4, steering with $\mathbf{v}_{\text{vis}}^{\perp(\ell)}$ significantly reduces hallucination rates while preserving general reasoning capabilities, overcoming the trade-off observed with the naive vector $\mathbf{v}_{\text{raw}}^{(\ell)}$. To further investigate this structural robustness, we conduct a continuous steering experiment by sweeping the injection intensity $\alpha$. Unlike $\mathbf{v}_{\text{raw}}^{(\ell)}$, which causes model collapse such as infinite repetition at high magnitudes due to feature entanglement, $\mathbf{v}_{\text{vis}}^{\perp(\ell)}$ maintains generation stability. Additionally, we observe that at extreme intensities, the model tends to generate visual placeholders (*e.g.*, `"image.jpg"`), a phenomenon further detailed in Appendix E.3.

### 3.3. Motivation for Sparse Intervention

Visual grounding is a layer-specific property. As shown in our latent space analysis (Appendix D), the separability between factual and hallucinated states is absent in shallow layers and only emerges deeper in the network. Consequently, dense multi-layer intervention is redundant; a single, targeted injection at an optimal depth $\ell^*$ suffices. We identify this layer by seeking robust geometric alignment, specifically where the steering vector favors the factual state over the hallucinated one: $\mathbf{h}_{\text{gt}}^{(\ell)} \cdot \mathbf{v}_{\text{vis}}^{\perp(\ell)} > \mathbf{h}_{\text{hall}}^{(\ell)} \cdot \mathbf{v}_{\text{vis}}^{\perp(\ell)}$. This condition ensures the intervention effectively discriminates against hallucinations, directly motivating the **Layer Selection** in Section 4.2.

## 4. REVIS: Design Details

Guided by the insights established in Section 3, we propose REVIS, a framework that explicitly decouples visual semantics from language priors to mitigate hallucinations. As shown in Figure 4, the design proceeds in three stages: orthogonal projection to extract a purified visual vector, calibration-based selection to identify the optimal layer, and dynamic steering mechanism during inference.

### 4.1. Orthogonal Visual Vector Construction

We synthesize the purified visual vector $\mathbf{v}_{\text{vis}}^{\perp(\ell)}$ by strictly isolating visual semantics from the model's language priors at each layer $\ell$. To achieve this, we utilize an extraction set $\mathcal{D}_{\text{ext}}$ ($N = 100$ paired examples randomly derived from

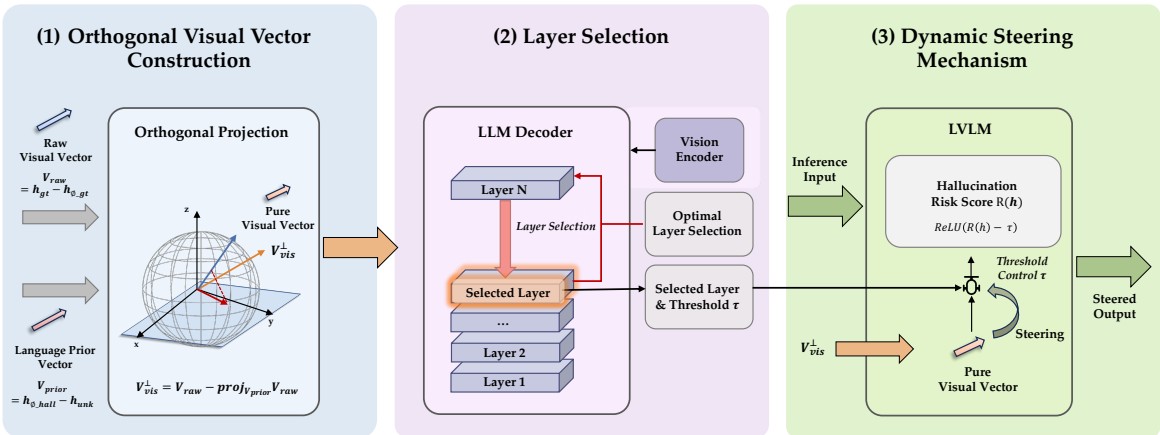

**Figure 4.** Design of REVIS. REVIS utilizes orthogonal projection to extract purified visual vectors, and performs sparse intervention through calibration-based layer selection and inference-time dynamic risk-aware steering.

Nullu (Yang et al., 2025b)) to compute the foundational vectors.

Using the raw visual vector $\mathbf{v}_{\text{raw}}^{(\ell)}$ and the language prior vector $\mathbf{v}_{\text{prior}}^{(\ell)}$ (calculated over $\mathcal{D}_{\text{ext}}$), we apply the Gram-Schmidt process to project $\mathbf{v}_{\text{raw}}^{(\ell)}$ onto the orthogonal complement of $\mathbf{v}_{\text{prior}}^{(\ell)}$:

$$
\begin{aligned}
\mathbf{v}_{\text{vis}}^{\perp(\ell)} &= \mathbf{v}_{\text{raw}}^{(\ell)} - \text{proj}_{\mathbf{v}_{\text{prior}}^{(\ell)}} \mathbf{v}_{\text{raw}}^{(\ell)} \\
&= \mathbf{v}_{\text{raw}}^{(\ell)} - \frac{\mathbf{v}_{\text{raw}}^{(\ell)} \cdot \mathbf{v}_{\text{prior}}^{(\ell)}}{\|\mathbf{v}_{\text{prior}}^{(\ell)}\|^2} \mathbf{v}_{\text{prior}}^{(\ell)}
\end{aligned}
\quad (3)
$$

This operation guarantees $\mathbf{v}_{\text{vis}}^{\perp(\ell)} \perp \mathbf{v}_{\text{prior}}^{(\ell)}$. By construction, steering along this vector injects visual information without projecting magnitude onto the pure fabrication manifold, effectively resolving the semantic collapse observed in naive approaches.

### 4.2. Layer Selection via Calibration

Guided by the sparsity analysis in Section 3.3, we identify the optimal intervention layer $L^*$ using a calibration-based search.

**Calibration Setup.** We construct a calibration dataset $\mathcal{D}_{\text{cal}}$ comprising $N = 100$ images from COCO train 2014 (Lin et al., 2014) using a POPE-like protocol (Li et al., 2023b). For each image, we formulate binary existence queries and extract the hidden states from correct responses (factual) and incorrect responses (hallucinated). This yields two state sets for each layer: $\mathcal{H}_{\text{fact}}^{(\ell)}$ and $\mathcal{H}_{\text{hall}}^{(\ell)}$.

**Layer Seclection.** To satisfy the geometric consistency $\mathbf{h}_{\text{gt}} \cdot \mathbf{v}_{\text{vis}}^{\perp} > \mathbf{h}_{\text{hall}} \cdot \mathbf{v}_{\text{vis}}^{\perp}$ established in Section 3.3, we seek the deepest semantic layer where the two distributions are separable. Given hidden state $\mathbf{h}$, we define its hallucination

risk score as $R(\mathbf{h}) = -\text{CosSim}(\mathbf{h}, \mathbf{v}_{\text{vis}}^{\perp(\ell)})$. We slightly abuse the notation and define $R(\mathcal{H}) = \frac{1}{|\mathcal{H}|} \sum_{h \in \mathcal{H}} R(\mathbf{h})$ as the average score on the hidden state set $\mathcal{H}$. We employ a backward search order ($L \rightarrow 1$) to select the deepest layer $L^*$ that satisfies the positive separability constraint:

$$
L^* = \max \left\{ \ell \mid R\big(\mathcal{H}_{\text{hall}}^{(\ell)}\big) - R\big(\mathcal{H}_{\text{fact}}^{(\ell)}\big) > 0 \right\} \quad (4)
$$

This formulation justifies the backward search, as it halts at the first instance (from the top) where the visual vector effectively discriminates between grounded and ungrounded states.

**Intervention Boundary.** We define a safety threshold $\tau$ based on the factual set $\mathcal{S}_{\text{fact}} = \{R(\mathbf{h}) \mid \mathbf{h} \in \mathcal{H}_{\text{fact}}^{(L^*)}\}$. To limit false positives, we set:

$$
\tau = \text{Percentile}\,(\mathcal{S}_{\text{fact}},\ k) \quad (5)
$$

Here, $k$ is a model-specific hyperparameter that controls the intervention sensitivity. This boundary ensures intervention only occurs when the representation drifts beyond the typical risk profile of factual generations.

### 4.3. Dynamic Steering Mechanism

At inference time step $t$, we compute the instantaneous risk $R_t = R(\mathbf{h}_t^{(L^*)})$ and apply an indicator-based steering activation:

$$
\lambda(t) = \alpha \cdot \mathbb{1}(R_t > \tau) \quad (6)
$$

If $R_t \leq \tau$, the gate remains closed ($\lambda(t) = 0$), preserving model capability. If $R_t > \tau$, indicating a loss of visual grounding, we inject the correction vector with constant strength:

$$
\tilde{\mathbf{h}}_t^{(L^*)} = \mathbf{h}_t^{(L^*)} + \lambda(t) \cdot \mathbf{v}_{\text{vis}}^{\perp(L^*)} \quad (7)
$$

This mechanism dynamically aligns the latent state with visual evidence only when necessary.

**Algorithm 1** REVIS: Sparse Orthogonal Intervention
_______________________________________________

1: **Input:** LVLM $\mathcal{M}$, Data $\mathcal{D}_{\text{ext}}, \mathcal{D}_{\text{cal}}$, Hyperparams $\alpha, k$;
2: **Output: y**.

3: *// Phase 1: Orthogonal Visual Vector Construction*
4: Compute raw vectors $\mathbf{v}_{\text{raw}}^{(\ell)}$ and $\mathbf{v}_{\text{prior}}^{(\ell)}$ on $\mathcal{D}_{\text{ext}}$     $\forall \ell \in [1, L]$
5: $\mathbf{v}_{\text{vis}}^{\perp(\ell)} \leftarrow \mathbf{v}_{\text{raw}}^{(\ell)} - \text{Proj}_{\mathbf{v}_{\text{prior}}^{(\ell)}} (\mathbf{v}_{\text{raw}}^{(\ell)})$ {Gram-Schmidt} $\forall \ell \in [1, L]$

6: *// Phase 2: Layer Selection (Backward Search)*
7: **for** $\ell = L$ **down to** $1$ **do**
8:     $\Delta_\ell \leftarrow R(\mathcal{H}_{\text{hall}}^{(\ell)}) - R(\mathcal{H}_{\text{fact}}^{(\ell)})$ {Using $R(\mathbf{h}) = -\text{CosSim}(\mathbf{h}, \mathbf{v}_{\text{vis}}^{\perp(\ell)})$}
9:     **if** $\Delta_\ell > 0$ **then** $L^* \leftarrow \ell$; **break end if** {Deepest valid layer}
10: **end for**
11: $\tau \leftarrow \text{Percentile}(\{R(\mathbf{h}) \mid \mathbf{h} \in \mathcal{H}_{\text{fact}}^{(L^*)}\}, k)$

12: *// Phase 3: Dynamic Inference*
13: **while** not EOS **do**
14:     $\mathbf{h}_t \leftarrow \mathcal{M}.\text{encode}(x_{<t})$ at layer $L^*$
15:     $\tilde{\mathbf{h}}_t \leftarrow \mathbf{h}_t + \alpha \cdot \mathbb{1}(R_t - \tau) \cdot \mathbf{v}_{\text{vis}}^{\perp(L^*)}$ {Risk-Aware Intervention}
16:     $y_t \leftarrow \text{Generate}(\tilde{\mathbf{h}}_t, \mathcal{M})$
17: **end while**
_______________________________________________

# 5. Experiments

## 5.1. Setup

**Datasets & Baselines.** We conduct evaluations on five standard benchmarks: **POPE** (Li et al., 2023b), **CHAIR** (Rohrbach et al., 2018), **MME** (Yin et al., 2024), **MM-Vet** (Yu et al., 2023), and **MMMU-Pro** (Yue et al., 2025). These datasets assess a spectrum of capabilities from object existence to complex integrated reasoning. We compare REVIS against five state-of-the-art methods: VCD (Leng et al., 2024), M3ID (Favero et al., 2024), ONLY (Wan et al., 2025), AGLA (An et al., 2025), and VTI (Liu et al., 2024c). Detailed configurations are provided in Appendix A and B.

**Implementation.** We implement REVIS following the core procedure summarized in Algorithm 1, with full details provided in Algorithm 2 in Appendix G. Experiments are performed on Qwen2.5-VL-7B-Instruct (Bai et al., 2025b), Qwen2.5-VL-32B-Instruct (Bai et al., 2025b), Qwen3-VL-8B-Instruct (Bai et al., 2025a), LLaVA-1.5-7B (Liu et al., 2023), LLaVA-NeXT-7B (Mistral-based) (Liu et al., 2024a), InternVL3-8B (Zhu et al., 2025), and InternVL3.5-8B (Wang et al., 2025). For readability, subsequent mentions omit parameter sizes for the default small variants and explicitly mark only the 32B model. For REVIS, steering vectors are computed using a dataset ($\mathcal{D}_{\text{ext}}$) of $N = 100$ paired examples from Nullu (Yang et al., 2025b). Unless otherwise stated, we set $k = 0.8$ and $\alpha = 1.6$ for Qwen2.5-VL-7B-Instruct. More details in Appendix C.3.

## 5.2. Overall Performance

We evaluate REVIS on Qwen2.5-VL to assess the effectiveness of REVIS. As shown in Table 3, REVIS consistently outperforms all baseline methods. A case study of hallucination mitigation is further provided in Appendix E.1, where we illustrate how REVIS mitigates complex hallucinations (*e.g.*, correcting attribute errors).

**Results on POPE & CHAIR.** REVIS effectively mitigates object hallucinations across both discriminative and generative tasks. In POPE, it achieves the highest F1 scores across all subsets. Notably, in the *Adversarial* setting, REVIS surpasses the strongest baseline (VCD) by 0.77% in Accuracy. For the generative task, REVIS reduces the CHAIR$_S$ score of the Regular baseline from 31.00% to 25.00%. This corresponds to a substantial **19.35% reduction** in hallucination rate, significantly outperforming logit-calibration methods like AGLA (29.00%) and ONLY (33.00%), which struggle to correct internal semantic drifts. Furthermore, results under varying maximum token settings (Appendix F.6) confirm REVIS's robustness.

**Results on MME, MM-Vet & MMMU-Pro.** Crucially, the aforementioned safety gains do not come at the cost of reasoning capabilities. While intervention-based baselines like VTI suffer from catastrophic degradation in utility (*e.g.*, MM-Vet score plummets from 70.18 to 56.38), REVIS maintains or even enhances general performance. Specifically, REVIS elevates the MME Overall score to 2345.30, marking a clear improvement over the Regular baseline's score of 2327.52. Furthermore, it achieves an MM-Vet score of 72.16, effectively surpassing the original model. On MMMU-Pro, REVIS improves the average score of Qwen2.5-VL from 0.348 to 0.357 (Appendix F.3). This confirms that our orthogonal projection successfully filters visual noise without disrupting the essential language priors required for complex reasoning.

## 5.3. Generalization to More Architectures

To verify generalization, we extend our evaluation to LLaVA-NeXT and LLaVA-1.5 architectures. We also validate REVIS on the more recent Qwen3-VL, InternVL3, and InternVL3.5 models (results in Appendix F.4), where the method maintains its effectiveness in mitigating hallucinations. Comprehensive results for the LLaVA series are presented in Table 5 and Table 9 in Appendix F.1. These experiments demonstrate that REVIS works consistently across diverse visual encoders and LLM backbones.

**Results on LLaVA-NeXT.** REVIS demonstrates superior robustness on this advanced baseline. It achieves the best performance on hallucination benchmarks, recording the lowest CHAIR$_S$ score (26.00%) and the highest F1 score (86.73%) on the challenging POPE-Adversarial subset, indicating

*Table 3.* Performance comparison on POPE and CHAIR benchmarks. **POPE** metrics (Accuracy, Recall, F1) are reported in percentage across three settings (↑ indicates higher is better). **CHAIR** metrics evaluate hallucination rates (↓ indicates lower is better).

| Method | POPE - *Random* (%) | | | POPE - *Popular* (%) | | | POPE - *Adversarial* (%) | | | CHAIR[†] (%) | |
| --- | --- | --- | --- | --- | --- | --- | --- | --- | --- | --- | --- |
| | ACC ↑ | Rec. ↑ | F1 ↑ | ACC ↑ | Rec. ↑ | F1 ↑ | ACC ↑ | Rec. ↑ | F1 ↑ | $C_S$ ↓ | $C_I$ ↓ |
| Regular[‡] | 89.27 | 79.73 | 88.14 | 88.23 | 79.73 | 87.14 | 87.10 | 79.87 | 86.09 | 31.00 | 8.13 |
| VCD | 89.60 | 80.93 | 88.61 | 88.27 | 80.93 | 87.34 | 87.03 | 80.93 | 86.19 | 34.00 | 9.06 |
| M3ID | 89.27 | 79.73 | 88.14 | 88.23 | 79.73 | 87.14 | 87.10 | 79.87 | 86.09 | 31.00 | 8.88 |
| Only | 89.33 | 80.20 | 88.26 | 88.37 | 80.20 | 87.33 | 86.93 | 80.13 | 85.98 | 33.00 | 9.75 |
| AGLA | 89.10 | 79.87 | 87.92 | 88.03 | 79.33 | 86.89 | 86.73 | 79.27 | 85.67 | 29.00 | 8.26 |
| VTI | 89.40 | 80.40 | 88.35 | 87.57 | 80.40 | 86.61 | 85.27 | 80.40 | 84.51 | 35.00 | 7.49 |
| **Ours** | 91.90 | 86.40 | 91.43 | 90.30 | 86.40 | 89.91 | 87.80 | 86.40 | 87.63 | 25.00 | 8.23 |

†: For CHAIR metrics ($C_S$ denotes CHAIR$_S$, $C_I$ denotes CHAIR$_I$), results are evaluated with *max_new_token*=512.
‡: Regular denotes the standard greedy decoding baseline.

*Table 4.* Quantitative comparison on MME and MM-Vet benchmarks. **MME** measures perception and cognition capabilities (higher scores indicate better performance). **MM-Vet** evaluates integrated problem-solving skills across 6 core capabilities. Best results are highlighted in **bold**.

| Method | MME (↑) | | | MM-Vet (↑) | | | | | | |
| --- | --- | --- | --- | --- | --- | --- | --- | --- | --- | --- |
| | Perc. | Cog. | Overall | Rec | OCR | Know | Gen | Spat | Math | **Total** |
| Regular[‡] | 1715.73 | 611.79 | 2327.52 | 65.67 | 76.67 | 58.57 | 59.00 | 70.40 | 72.69 | 70.18 |
| VCD | 1665.65 | 593.21 | 2258.86 | 64.80 | 73.96 | 57.74 | 60.12 | 66.27 | 72.69 | 68.76 |
| M3ID | 1715.73 | 611.79 | 2327.52 | 64.83 | 76.77 | 58.99 | 61.50 | 71.07 | 76.54 | 69.43 |
| Only | 1686.78 | 603.21 | 2289.99 | 64.87 | 77.92 | 56.90 | 59.50 | 71.60 | 76.54 | 70.05 |
| AGLA | 1716.80 | 611.79 | 2328.59 | 65.03 | 76.77 | 58.63 | 60.38 | 68.40 | 72.69 | 69.56 |
| VTI | 1492.39 | 591.43 | 2083.82 | 48.53 | 66.87 | 39.88 | 37.75 | 63.07 | 76.92 | 56.38 |
| **Ours** | 1717.44 | 627.86 | 2345.30 | 67.93 | 78.85 | 61.55 | 61.37 | 72.53 | 84.62 | 72.16 |

*Note*: For MME, Perc. denotes Perception and Cog. denotes Cognition. For MM-Vet, categories are Recognition (Rec), OCR, Knowledge (Know), Generation (Gen), Spatial Awareness (Spat), and Math.
‡: Regular denotes the standard greedy decoding baseline.

*Table 5.* Performance comparison on POPE and CHAIR benchmarks across LLaVA-1.5 and LLaVA-NeXT architectures. **POPE** metrics are reported in percentage (↑ higher is better). **CHAIR** metrics evaluate hallucination rates (↓ lower is better).

| Model | Method | POPE - *Random* (%) (↑) | | | POPE - *Popular* (%) (↑) | | | POPE - *Adversarial* (%) (↑) | | | CHAIR[†] (%) (↓) | |
| --- | --- | --- | --- | --- | --- | --- | --- | --- | --- | --- | --- | --- |
| | | ACC | Rec | F1 | ACC | Rec | F1 | ACC | Rec | F1 | $C_S$ | $C_I$ |
| **LLaVA-1.5** | Regular[‡] | 87.67 | 78.40 | 86.41 | 85.90 | 78.40 | 84.76 | 83.20 | 78.20 | 82.32 | 54.00 | 15.78 |
| | VCD | 86.97 | 79.00 | 85.84 | 84.27 | 79.00 | 83.39 | 81.23 | 78.93 | 80.79 | 52.00 | 16.13 |
| | M3ID | 87.67 | 78.40 | 86.41 | 85.90 | 78.40 | 84.76 | 83.20 | 78.20 | 82.32 | 49.00 | 13.78 |
| | ONLY | 87.03 | 78.60 | 85.84 | 85.07 | 78.60 | 84.03 | 81.83 | 78.53 | 81.21 | 50.00 | 14.03 |
| | AGLA | 87.67 | 78.53 | 86.43 | 85.90 | 78.53 | 84.78 | 83.23 | 78.33 | 82.37 | 51.00 | 13.64 |
| | VTI | 87.57 | 77.47 | 86.17 | 85.23 | 77.47 | 83.99 | 81.77 | 77.27 | 80.91 | 35.00 | 15.58 |
| | **Ours** | 89.37 | 83.33 | 88.68 | 87.13 | 83.33 | 86.63 | 83.50 | 83.20 | 83.51 | 30.00 | 14.16 |
| **LLaVA-NeXT** | Regular[‡] | 89.90 | 81.20 | 88.94 | 88.67 | 81.20 | 87.75 | 86.87 | 81.20 | 86.08 | 30.00 | 7.49 |
| | VCD | 89.83 | 82.13 | 88.99 | 88.30 | 82.13 | 87.53 | 86.13 | 82.07 | 85.55 | 33.00 | 8.06 |
| | M3ID | 89.90 | 81.20 | 88.94 | 88.67 | 81.20 | 87.75 | 86.87 | 81.20 | 86.08 | 32.00 | 8.38 |
| | ONLY | 89.97 | 81.93 | 89.09 | 88.60 | 81.93 | 87.79 | 86.13 | 81.87 | 85.52 | 30.00 | 7.19 |
| | AGLA | 89.93 | 81.20 | 88.97 | 88.67 | 81.20 | 87.75 | 86.87 | 81.20 | 86.08 | 30.00 | 7.88 |
| | VTI | 89.77 | 81.60 | 88.86 | 88.90 | 81.60 | 88.03 | 86.57 | 81.60 | 85.86 | 29.00 | 6.10 |
| | **Ours** | 90.70 | 84.07 | 90.04 | 89.57 | 84.07 | 88.96 | 87.13 | 84.07 | 86.73 | 26.00 | 6.32 |

†: For CHAIR metrics ($C_S$ denotes CHAIR$_S$, $C_I$ denotes CHAIR$_I$), results are evaluated with *max_new_token*=512.
‡: Regular decoding denotes the standard greedy decoding baseline.

strong resistance to misleading visual queries. Crucially, regarding utility, REVIS increases the MME Overall score to 1817.32 (surpassing the baseline's 1803.98) while maintaining a competitive MM-Vet score of 47.48. On MMMU-Pro, REVIS preserves the LLaVA-NeXT average score (0.248 vs. 0.248; Appendix F.3). This confirms that our intervention mitigates object hallucinations without compromising the model's complex reasoning capabilities.

**Results on LLaVA-1.5.** The improvements are particularly pronounced on this architecture. REVIS yields significant gains across all metrics, securing the highest accuracy across all three POPE subsets (Random, Popular, and Adversar-

ial). Notably, it reduces the CHAIR$_S$ score from 54.00% to 30.00%, corresponding to a substantial 44.4% reduction, which effectively suppresses the model's intrinsic tendency to fabricate objects. Furthermore, REVIS outperforms the baseline on MME (1787.27 vs. 1754.33) and maintains a stable MM-Vet score of 33.53, validating that visual alignment correction does not degrade general knowledge.

## 5.4. Scalability to a Larger-Scale Backbone

We evaluate REVIS on Qwen2.5-VL-32B-Instruct to assess its scalability to larger-scale backbones. As detailed in Appendix F.3, REVIS reduces CHAIR$_S$ across 64, 128,

*Table 6.* Ablation study on $\alpha$. The average POPE metrics (Random, Popular, Adversarial) and CHAIR metrics.

| $\alpha$ | POPE (%) | | | CHAIR (%) | | MM-Vet ↑ |
|---|---|---|---|---|---|---|
| | Acc ↑ | Rec. ↑ | F1 ↑ | $C_S$ ↓ | $C_I$ ↓ | |
| Regular | 88.20 | 79.78 | 87.12 | 31.00 | 8.13 | 70.18 |
| 1.0 | 89.58 | 84.27 | 89.01 | 27.00 | 7.28 | 73.12 |
| 1.2 | 89.74 | 84.93 | 89.25 | 26.00 | 6.62 | 72.11 |
| 1.4 | 89.89 | 85.73 | 89.47 | 29.00 | 7.33 | 71.06 |
| 1.6 | 90.00 | 86.40 | 89.66 | 25.00 | 8.23 | 72.16 |
| 1.8 | 90.07 | 87.27 | 89.81 | 30.00 | 8.90 | 69.45 |
| 2.0 | 89.70 | 87.40 | 89.50 | 29.00 | 8.33 | 69.95 |

*Table 7.* Ablation study on different layers. $\alpha = 1.6$ is used for all layers.

| Layer | POPE (%) | | | CHAIR (%) | | MM-Vet ↑ |
|---|---|---|---|---|---|---|
| | Acc ↑ | Rec. ↑ | F1 ↑ | $C_S$ ↓ | $C_I$ ↓ | |
| Regular | 88.20 | 79.78 | 87.12 | 31.00 | 8.13 | 70.18 |
| 23 | 81.88 | 64.80 | 78.15 | 37.00 | 9.75 | 60.37 |
| 24 | 87.57 | 77.91 | 86.24 | 31.00 | 8.59 | 67.98 |
| 25 | 85.51 | 72.98 | 83.44 | 29.00 | 9.23 | 68.58 |
| 26 | 86.07 | 74.42 | 84.24 | 32.00 | 9.72 | 68.94 |
| 27 | 90.00 | 86.40 | 89.66 | 25.00 | 8.23 | 72.16 |

and 512 maximum-token budgets (9.00/22.00/52.00 → 8.00/18.00/46.00), while preserving MM-Vet performance (74.31 vs. 74.13). On MMMU-Pro, REVIS improves the Qwen2.5-VL-32B-Instruct average score from 0.422 to 0.435, with full results provided in Appendix F.3. These results indicate that sparse latent steering scales to larger models without sacrificing general multimodal utility.

## 5.5. Ablation Study

In this section, we conduct ablation studies to validate the effectiveness of our core design components. Specifically, we investigate the impact of steering intensity ($\alpha$), adaptive layer selection, and the dynamic threshold ($\tau$).

### 5.5.1. EFFECT OF HYPER-PARAMETERS

Table 6 shows that hallucination mitigation follows a convex trajectory relative to $\alpha$, reaching an optimum at $\alpha = 1.6$. Beyond this point, $\text{CHAIR}_S$ degrades ($25.00\% \rightarrow 29.00\%$), indicating that excessive projection introduces high-variance noise. Simultaneously, general reasoning capabilities remain robust; the MM-Vet score at $\alpha = 1.6$ (72.16) surpasses the Regular baseline (70.18). Thus, a balanced $\alpha$ is critical for effective steering without compromising other capabilities.

### 5.5.2. EFFECT OF LAYER SELECTION

We compare Adaptive Layer Selection against fixed-layer interventions (Table 7). Intervention at shallow layers (*e.g.*, Layer 23) is ineffective ($\text{CHAIR}_S$ 37.00%) and significantly harms utility (MM-Vet drops to 60.37), confirming that visual semantics are entangled early on. Performance peaks at

*Table 8.* The adaptive threshold $\tau$ is critical for preventing model collapse and preserving precision.

| Model | Setting | POPE (%) | CHAIR (%) | | MM-Vet |
|---|---|---|---|---|---|
| | | Avg. ↑ | $C_S$ ↓ | $C_I$ ↓ | |
| Qwen2.5-VL | w/o $\tau$ | 90.02 | 25.0 | 8.1 | 72.57 |
| | Ours | 90.00 | 25.0 | 8.2 | 72.16 |
| LLaVA-NeXT | w/o $\tau$ | 88.93 | 34.0 | 9.9 | 46.70 |
| | Ours | 89.13 | 26.0 | 6.3 | 47.48 |
| LLaVA-1.5 | w/o $\tau$ | 86.50 | ∅† | ∅† | 28.81 |
| | Ours | 86.67 | 30.0 | 14.2 | 33.53 |

†: Indicates *Model Collapse* (the model generates infinite repetition loops and fails to produce valid outputs).

Layer 27 for Qwen2.5-VL. Notably, Layer 27 also yields the highest MM-Vet score (72.16), suggesting that the optimal depth for hallucination mitigation coincides with the layer richest in semantic reasoning. Crucially, results on LLaVA-1.5 and LLaVA-NeXT (Appendix F.2) confirm that $L^*$ is model-specific, necessitating our calibration-based search.

### 5.5.3. EFFECT OF THRESHOLD ($\tau$)

Table 8 demonstrates the necessity of the dynamic risk threshold. Removing $\tau$ (static steering) causes *Model Collapse* in sensitive models like LLaVA-1.5 (detailed in Appendix E.2) and degrades $\text{CHAIR}_S$ ($34.00\% \rightarrow 26.00\%$) in robust models. This instability extends to general reasoning capabilities; removing $\tau$ leads to a sharp decline in metrics like MM-Vet on LLaVA-1.5, confirming that unconditional steering disrupts the model's generation process. This proves that REVIS must operate as a precision filter to ensure generation stability.

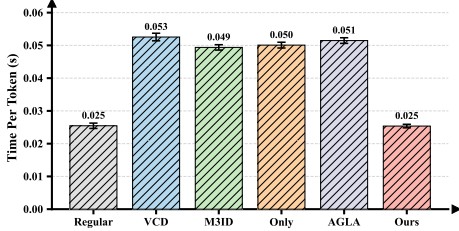

*Figure 5.* Inference Latency Comparison. We report the mean Time Per Token (TPT) in seconds. REVIS maintains the efficiency of Regular decoding.

## 5.6. Inference Latency

We evaluate computational efficiency in Figure 5. REVIS achieves a mean Time Per Token (TPT) of 0.025s, statistically indistinguishable from Regular decoding ($\sim$0.025s). In contrast, contrastive methods like VCD and M3ID double the inference cost ($\sim$0.05s) due to required parallel forward passes. REVIS thus incurs negligible overhead while delivering superior mitigation, making it highly suitable for latency-constrained practical deployments.

# 6. Conclusion

In this paper, we present REVIS, a training-free framework that employs sparse latent steering to mitigate object hallucination in Large Vision-Language Models. By decoupling semantics of visual information via orthogonal projection, REVIS effectively restores grounding without compromising general reasoning capabilities. Extensive experiments confirm its superior mitigation performance, inference efficiency, and broad applicability across diverse architectures.

# 7. Limitations & Future Work

Similar to other inference-time methods, REVIS relies on the base model's representations, limiting its ability to rectify absolute perceptual blindness. Additionally, our evaluation focuses on 7B/8B and 32B architectures; verifying scalability on larger models (*e.g.*, 70B+) remains for future work.

## Impact Statement

This paper presents work whose goal is to advance the field of Machine Learning, specifically in improving the reliability of Vision-Language Models (VLMs) by mitigating hallucinations. There are many potential societal consequences of our work, none which we feel must be specifically highlighted here.

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

# A. Dataset Details

To comprehensively evaluate hallucination mitigation and general capabilities, we employ four standard benchmarks.

**CHAIR (Rohrbach et al., 2018).** This benchmark assesses the alignment between generated captions and image content. We utilize the val2014 split of the MSCOCO dataset (Lin et al., 2014), randomly selecting 100 images for evaluation. The model is prompted with "Please describe this image in detail." Performance is quantified using two metrics: $\text{CHAIR}_I$ (object-level) and $\text{CHAIR}_S$ (sentence-level), defined as:

$$\text{CHAIR}_I = \frac{|\{\text{hallucinated objects}\}|}{|\{\text{all objects mentioned}\}|}, \quad \text{CHAIR}_S = \frac{|\{\text{sentences with hallucinated objects}\}|}{|\{\text{all sentences}\}|} \tag{8}$$

**POPE (Li et al., 2023b).** Polling-based Object Probing Evaluation (POPE) evaluates object existence via a binary VQA format. We use the specific prompt: "Is [object] in this image? Please answer yes or no." To test robustness against statistical biases, we report performance across three sampling settings: *Random*, *Popular*, and *Adversarial*.

**MME (Yin et al., 2024).** MME measures both Perception and Cognition capabilities using concise Yes/No queries. It covers 14 subtasks ranging from fine-grained existence and count detection to complex commonsense reasoning. We report the aggregate scores for both Perception and Cognition subsets.

**MM-Vet (Yu et al., 2023).** This benchmark evaluates integrated multimodal capabilities across six core domains, including knowledge, OCR, and spatial awareness. It relies on an LLM-based scorer to assess open-ended responses. Note that for this evaluation, we utilize **Qwen3-235B-A22B-Instruct-2507** (Yang et al., 2025a) as the judge model.

**MMMU-Pro (Yue et al., 2025).** This benchmark extends MMMU with more robust evaluation variants, including expanded multiple-choice options and a vision-only variant where textual context is embedded in the image. We report the 4-option, 10-option, vision-only, and averaged scores.

# B. Baseline Details

We compare our proposed method against five state-of-the-art baselines. VCD (Leng et al., 2024) employs contrastive decoding to penalize language priors by contrasting logits from original and masked visual inputs. M3ID (Favero et al., 2024) maximizes the mutual information between visual inputs and generated text to enhance grounding. ONLY (Wan et al., 2025) utilizes a training-free decoding strategy that identifies attention heads with high text-to-visual entropy ratios and applies a single-layer intervention to reduce language bias without requiring multiple queries. AGLA (An et al., 2025) mitigates hallucinations by assembling global features with prompt-relevant local features; it generates an augmented view via image-prompt matching and calibrates the logit distribution to highlight discriminative visual cues. VTI (Liu et al., 2024c) utilizes a steering-based approach, intervening in both the visual encoder and text decoder using vectors derived from contrastive instruction tuning data.

# C. Implementation Details

## C.1. Analysis & Steering Dataset Construction Protocol

Following the methodology established in Nullu (Yang et al., 2025b), we construct paired vision-language inputs to isolate the mechanistic roots of hallucination. Specifically, for each image sample, we pair it with two distinct textual descriptions: a ground-truth description that accurately depicts the scene, and a hallucinated description containing non-existent objects. These hallucinated captions are generated by modifying the ground-truth descriptions using GPT-3.5, guided by factors such as object co-occurrence and uncertainty to ensure plausibility. This pairing strategy ensures that the difference between factual and hallucinatory states stems solely from the truthfulness of object tokens rather than syntactic variations, providing a rigorous basis for extracting steering vectors.

## C.2. Refusal Response Construction

Insight from the work (Kalai et al., 2025), we want to induce the model to say "I don't know" when the visual input is none. To construct the neutral refusal state ($\mathbf{h}_{\emptyset\_\text{unk}}$) used in calculating the language prior vector, we utilize a randomized pool of responses rather than a single fixed template. This approach avoids overfitting to specific lexical artifacts and ensures the vector represents a generalized state of ignorance. The standardized pool is defined in the toolbox below:

**Refusal Response Pool**

- "I cannot answer this question because there is no image provided."
- "Without an image, I cannot describe the scene."
- "Please provide an image so I can describe it."
- "I don't see any image here."
- "The visual information is missing."

During vector extraction, we average the hidden states across these responses to obtain a robust representation of the unknown state.

### C.3. Experimental Setup and Hyperparameters

We implement our framework using PyTorch on NVIDIA A100 GPUs. Regarding the hyper-parameters for the LLaVA family, we adopt a unified configuration to ensure consistency. For LLaVA-1.5, we set the steering intensity $\alpha$ to 1.1 and select the penultimate layer (Layer 31) for intervention. Similarly, for LLaVA-NeXT, we maintain the steering intensity at $\alpha = 1.1$ and apply the intervention at the penultimate layer (Layer 31).

## D. Extended Analysis of Latent Space

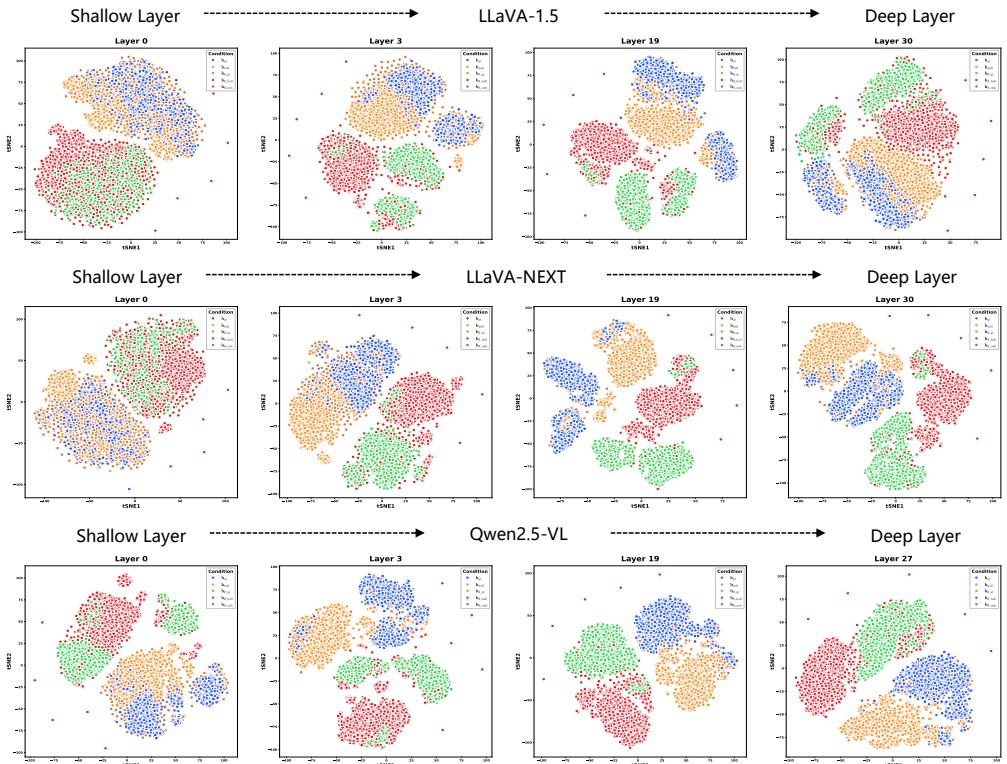

*Figure 6.* Layer-wise t-SNE visualization of latent states across LLaVA-1.5, LLaVA-NeXT, and Qwen2.5-VL. While LLaVA models exhibit significant entanglement in shallow layers, the more capable Qwen2.5-VL demonstrates earlier separability, indicating superior representational disentanglement.

In this section, we provide a detailed comparative visualization of the latent space evolution across three distinct architectures: LLaVA-1.5, LLaVA-NeXT, and Qwen2.5-VL. Figure 4 plots the t-SNE projections of the [EOS] token hidden states under the five counterfactual conditions described in the main text, tracking their progression from shallow to deep layers.

We observe a significant correlation between model capability and the geometric separability of these states. For the LLaVA

family (LLaVA-1.5 and LLaVA-NeXT), the five clusters remain highly entangled in the shallow layers, with clear linear separability emerging only in the deep layers. This aligns with our observation in §3.3 that visual semantics in these models are a late-stage emergent property.

In contrast, Qwen2.5-VL exhibits a distinct geometric pattern, where the clusters demonstrate notable separability even in shallower layers. We attribute this phenomenon to the superior representational capacity of Qwen2.5-VL. As a stronger model, it appears to possess a more disentangled latent space, capable of encoding the semantic distinction between factual visual perception and language priors more robustly and at earlier network depths. This suggests that while the functional "visual information" is generally concentrated in deeper layers for steering purposes, the intrinsic separability of representations is accelerated in architectures with higher capability.

# E. Case Study

## E.1. Analysis of Hallucination Mitigation

In this part, we present a qualitative comparison to demonstrate the effectiveness of our method in mitigating hallucinations. We compare our approach against the baseline Qwen2.5-VL without any intervention. The parameter of max new token is 512.

As illustrated in Case 1 below, the baseline model suffers from *attribute binding errors*, incorrectly conflating visual features of adjacent objects. Specifically, it hallucinates that the adult cutting the ribbon is wearing a helmet, likely confusing the attribute of the child standing nearby. In contrast, our method correctly disentangles these visual entities, accurately attributing the helmet to the child while describing the adult's attire correctly.

---

**Case 1: Mitigation of Attribute Hallucination**

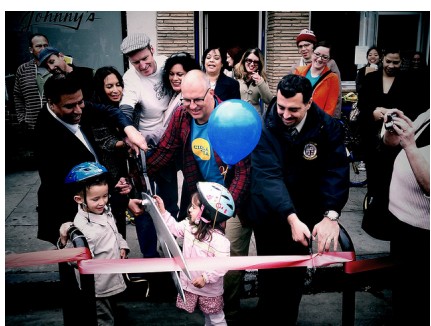

*Figure 7.* Visual Input

**Query:** Describe this image in detail.

**Qwen2.5-VL (Baseline):**
**Response:** This image captures a celebratory moment where a ribbon-cutting ceremony is taking place, likely to mark the opening of a new establishment or event. The scene is outdoors, with a group of people gathered around a table where a red ribbon is stretched across it. In the center of the image, two individuals are cutting the ribbon with scissors. One person on the left appears to be wearing a suit and a helmet, possibly indicating a role related to safety or construction. The individual on the right is dressed in a uniform, suggesting they might be part of an organization or government agency. A child in a pink jacket and helmet stands next to the adult in the suit, seemingly participating in the ceremony. Another child, also wearing a helmet, is positioned behind the ribbon, looking on. The child in the pink jacket seems to be holding a blue balloon, adding a festive touch to the occasion. The background shows a crowd of onlookers, some smiling and others observing the event. The setting appears to be a storefront or public space, as indicated by the brick wall and signage visible in the upper left corner. The atmosphere is lively and celebratory, with people dressed casually, suggesting a community-oriented event.

**Ours:**
**Response:** The image shows a group of people gathered outdoors, participating in what appears to be the ribbon-cutting ceremony for an event or establishment. The focus is on two individuals, one dressed in a navy-blue uniform and another in casual attire, who are cutting a red ribbon with scissors. A child, wearing a helmet, stands to the left, seemingly assisting or observing, while another child, also in a helmet, stands nearby, looking at the ribbon. The scene is lively, with several people in the background, some smiling, others engaged in conversation, and some capturing the moment with cameras. A blue balloon is visible, adding to the festive atmosphere. The setting seems to be outside a building, as indicated by the brick wall and doorway in the background. The mood appears celebratory, suggesting the opening of a new business, community center, or similar venue. The crowd includes both adults and children, contributing to the sense of community involvement.

---

### E.2. Necessity of Sparse Intervention

In this part, we conduct experiment on LLaVA-1.5 and demonstrate the necessity of the risk-calibrated strategy (Adaptive Layer Selection) discussed in Section 3.3. We validate that dense intervention (without threshold) disrupts the language manifold, leading to structural instability.

As shown in Case 2 below, applying visual steering at every step (w/o Threshold) introduces excessive noise, causing the model to fall into a degenerative repetition loop.

---

**Case 2: Impact of Dynamic Threshold**

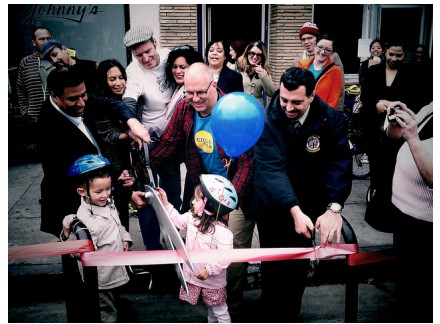

*Figure 8.* Visual Input

**Query:** Describe the image in detail.

**Ours (w/ Threshold):**
**Response:** The image features a group of people, including children, gathered for a ribbon-cutting ceremony. They are standing in front of a red ribbon, which is being cut by men. The ceremony is taking place outdoors, with some of the attendees wearing ties. Several children can be seen participating in the event, with some of them wearing ties. In addition, some of the attendees are wearing ties.

**Ours (w/o Threshold):**
**Response:** The image features a group of people, including children, gathered for a ribbon-cutting ceremony. They are standing in front of a red ribbon, which is being cut by an older man. The ceremony is taking place outdo **until until until until until until...**

---

### E.3. The "image.jpg" Phenomenon

In this part, we present an intriguing qualitative observation discussed in Section 3.2. To explore the semantic nature of our orthogonalized vector $\mathbf{v}_{vis}^{\perp}$, we conduct a steering experiment on LLaVA-1.5 by sweeping the injection intensity $\alpha$.

As shown in Case 3 below, distinct behaviors emerge across different intensities. At $\alpha = 0$ (Regular), the model outputs a generic description. Increasing the intensity to $\alpha = 1.1$ successfully reactivates suppressed visual details (*e.g.*, "yellow racquet", "white shirt"), indicating the vector's capacity to enhance visual grounding. Most notably, pushing the intensity to $\alpha = 1.5$ triggers the model to output `"image.jpg)"`. While not definitive proof, this phenomenon invites a fascinating hypothesis: $\mathbf{v}_{vis}^{\perp}$ may encode the **abstract concept** of visual modality itself. We speculate that when injected with excessive strength, this "meta-visual" signal overrides specific features, causing the model to collapse into a generic file-identifier representation.

---

**Case 3: The "image.jpg" Phenomenon**

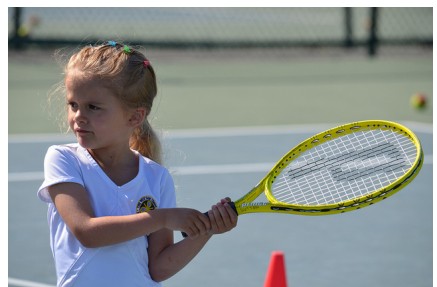

*Figure 9.* Visual Input

**Query:** Please describe this image in detail.

**Regular ($\alpha = 0$):**
**Response:** The image features a young girl holding a tennis racket, standing on a tennis court. She appears to be focused and ready to play. The tennis court is surrounded by a chain-link fence, providing a secure environment for the game.

**Ours ($\alpha = 1.1$):**
**Response:** In the image, a young girl is holding a yellow racquet, getting ready to play. She is wearing a white shirt and is focused on her game. She is standing on a court, which is equipped with cones for practice. One cone can be seen near her. Her racquet is prominently visible, as she grips it firmly.

**Ours ($\alpha = 1.5$):**
**Response:** image.jpg)

---

## F. More Results

### F.1. MME & MM-Vet Result on LLaVA-1.5 and LLaVA-NEXT

In this part, we report the utility performance on MME and MM-Vet benchmarks (Table 9). The results demonstrate that REVIS preserves the general reasoning capabilities of the original models, avoiding the significant performance degradation observed in baselines like VTI. Notably, we observe clear improvements in visual-centric tasks, such as **Spatial Awareness** ($26.93 \rightarrow 34.67$) and **Math** ($10.77 \rightarrow 14.62$) on LLaVA-1.5, confirming that reactivating visual evidence enhances precise grounding.

*Table 9.* Quantitative comparison on MME and MM-Vet benchmarks across different models. **MME** measures perception and cognition capabilities. **MM-Vet** evaluates integrated problem-solving skills. Best results are highlighted in **bold**.

| Model | Method | MME (↑) | | | MM-Vet (↑) | | | | | | |
| | | Perc. | Cog. | Overall | Rec | OCR | Know | Gen | Spat | Math | **Total** |
|---|---|---|---|---|---|---|---|---|---|---|---|
| LLaVA-1.5 | Regular‡ | 1455.42 | 298.93 | 1754.33 | 41.93 | 25.42 | 27.74 | 30.50 | 26.93 | 10.77 | 35.92 |
| | VCD | 1472.91 | 298.93 | 1771.84 | 42.20 | 25.00 | 28.81 | 30.62 | 27.20 | 5.77 | 36.10 |
| | M3ID | 1455.42 | 298.93 | 1754.33 | 42.87 | 25.94 | 25.12 | 28.25 | 30.27 | 11.54 | 36.74 |
| | Only | 1424.71 | 281.07 | 1705.78 | 43.13 | 25.00 | 27.74 | 30.87 | 26.80 | 3.85 | 36.15 |
| | AGLA | 1486.37 | 287.14 | 1773.51 | 42.20 | 25.10 | 28.45 | 30.87 | 26.53 | 10.77 | 36.01 |
| | VTI | 1240.31 | 267.14 | 1507.45 | 40.47 | 18.75 | 27.62 | 29.38 | 23.80 | 7.69 | 32.94 |
| | **Ours** | 1481.56 | 305.71 | 1787.27 | 36.07 | 29.79 | 25.12 | 28.00 | 34.67 | 14.62 | 33.53 |
| LLaVA-NEXT | Regular‡ | 1512.19 | 291.79 | 1803.98 | 49.00 | 42.29 | 41.79 | 44.12 | 40.53 | 25.00 | 46.97 |
| | VCD | 1463.05 | 291.07 | 1754.12 | 50.47 | 40.31 | 41.94 | 53.80 | 38.80 | 19.23 | 47.16 |
| | M3ID | 1512.19 | 291.79 | 1803.98 | 50.73 | 46.56 | 41.55 | 44.88 | 46.00 | 34.62 | 49.72 |
| | Only | 1510.88 | 298.57 | 1809.45 | 49.93 | 39.58 | 42.74 | 45.13 | 37.20 | 25.00 | 46.97 |
| | AGLA | 1511.94 | 291.79 | 1803.73 | 49.53 | 41.25 | 40.48 | 44.00 | 39.47 | 25.00 | 46.88 |
| | VTI | 1416.51 | 268.21 | 1684.72 | 46.73 | 36.25 | 40.12 | 42.88 | 35.73 | 15.77 | 43.44 |
| | **Ours** | 1514.46 | 302.86 | 1817.32 | 48.60 | 43.85 | 40.71 | 43.00 | 39.47 | 34.62 | 47.48 |

Note: For MME, Perc. denotes Perception and Cog. denotes Cognition. For MM-Vet, categories are Recognition (Rec), OCR, Knowledge (Know), Generation (Gen), Spatial Awareness (Spat), and Math.
‡: Regular denotes the standard greedy decoding baseline. AGLA results are currently pending.

### F.2. More Results of Layer Selection

In this part, we show the results of layer selection on LLaVA-NEXT. We report the averaged POPE metrics (Accuracy, Recall, and F1 score) across the Random, Popular, and Adversarial subsets, along with CHAIR and MM-Vet scores. As shown in Table 10, Layer 30 achieves the best trade-off.

*Table 10.* Ablation study on layer selection for LLaVA-NEXT. POPE results are averaged across three subsets.

| Layer | POPE (Avg) | | | CHAIR | | MM-Vet ↑ |
| | Acc ↑ | Rec ↑ | F1 ↑ | $C_S$ ↓ | $C_I$ ↓ | |
|---|---|---|---|---|---|---|
| 27 | 88.48 | 81.20 | 87.59 | 27.0 | 7.07 | 45.37 |
| 28 | 88.48 | 81.20 | 87.59 | 27.0 | 7.24 | 44.68 |
| 29 | 88.48 | 81.20 | 87.59 | 27.0 | 7.19 | 44.08 |
| 30 | 89.13 | 84.07 | 88.58 | 26.0 | 6.32 | 47.48 |
| 31 | 88.48 | 81.20 | 87.59 | 29.0 | 7.49 | 44.45 |

*Table 11.* Ablation study on layer selection for LLaVA-1.5. POPE results are averaged across three subsets.

| Layer | POPE (Avg) | | | CHAIR | | MM-Vet ↑ |
| | Acc ↑ | Rec ↑ | F1 ↑ | $C_S$ ↓ | $C_I$ ↓ | |
|---|---|---|---|---|---|---|
| 27 | 82.54 | 67.40 | 79.44 | ⊘† | ⊘† | 28.35 |
| 28 | 80.55 | 62.62 | 76.30 | 33.0 | 13.70 | 25.09 |
| 29 | 84.86 | 74.27 | 83.08 | 41.0 | 18.52 | 29.72 |
| 30 | 86.67 | 83.29 | 86.27 | 30.0 | 14.16 | 33.53 |
| 31 | 77.93 | 56.73 | 72.00 | 51.0 | 47.84 | 12.29 |

†: Indicates *Model Collapse* (the model generates infinite repetition loops and fails to produce valid outputs).

## F.3. Scalability and MMMU-Pro Evaluation

To evaluate whether REVIS scales beyond the default-scale setting, we conduct experiments on Qwen2.5-VL-32B-Instruct. For this larger model, calibration selects Layer 62, the third-to-last layer. As shown in Table 12, REVIS consistently reduces $CHAIR_S$ and $CHAIR_I$ across 64, 128, and 512 maximum-token budgets. The largest improvement appears in long-form generation, where $CHAIR_S$ drops from 52.00% to 46.00% and $CHAIR_I$ drops from 11.05% to 8.99%. At the same time, MM-Vet remains stable, indicating that the larger model preserves general multimodal reasoning under our intervention.

*Table 12.* Scalability results on Qwen2.5-VL-32B-Instruct. CHAIR is evaluated under different max-token budgets; MM-Vet and MMMU-Pro measure general multimodal reasoning.

| Method | Max Token = 64 | | Max Token = 128 | | Max Token = 512 | | MM-Vet ↑ | MMMU-Pro ↑ | | |
|---|---|---|---|---|---|---|---|---|---|---|
| | $C_S \downarrow$ | $C_I \downarrow$ | $C_S \downarrow$ | $C_I \downarrow$ | $C_S \downarrow$ | $C_I \downarrow$ | | 4-opt | 10-opt | Vision |
| Regular | 9.00 | 5.26 | 22.00 | 6.72 | 52.00 | 11.05 | **74.31** | 0.510 | 0.390 | 0.365 |
| **Ours** | **8.00** | **4.91** | **18.00** | **5.85** | **46.00** | **8.99** | 74.13 | **0.520** | **0.410** | **0.375** |

We further evaluate REVIS on MMMU-Pro to test whether hallucination mitigation affects benchmark performance. Table 13 shows that REVIS improves the average score on Qwen2.5-VL, Qwen2.5-VL-32B-Instruct, and Qwen3-VL, while preserving LLaVA-NeXT performance. These results support that REVIS primarily strengthens visual grounding rather than trading off general utility.

*Table 13.* MMMU-Pro results across model scales and architectures. We report accuracy in the 4-option, 10-option, and vision-only settings, together with their average.

| Method | Qwen2.5-VL | | | | Qwen2.5-VL-32B-Instruct | | | | Qwen3-VL | | | | LLaVA-NeXT | | | |
|---|---|---|---|---|---|---|---|---|---|---|---|---|---|---|---|---|
| | 4-opt | 10-opt | Vision | Avg. | 4-opt | 10-opt | Vision | Avg. | 4-opt | 10-opt | Vision | Avg. | 4-opt | 10-opt | Vision | Avg. |
| Regular | 0.410 | **0.335** | 0.300 | 0.348 | 0.510 | 0.390 | 0.365 | 0.422 | 0.460 | 0.370 | **0.310** | 0.380 | 0.415 | 0.195 | 0.135 | 0.248 |
| **Ours** | **0.420** | 0.315 | **0.335** | **0.357** | **0.520** | **0.410** | **0.375** | **0.435** | **0.480** | **0.400** | 0.295 | **0.392** | 0.415 | 0.195 | 0.135 | 0.248 |

## F.4. Generalization to Qwen3-VL and InternVL Series

To further verify the generalization capability of our framework, we extend the evaluation to Qwen3-VL and the InternVL family. As shown in Table 14, REVIS reduces hallucination on Qwen3-VL at longer generation lengths, lowering $CHAIR_S$ from 17.00% to 14.00% at 128 tokens and from 45.00% to 43.00% at 512 tokens. It also improves all POPE metrics, increasing Accuracy, Recall, and F1 to 89.48%, 85.69%, and 89.10%, respectively.

*Table 14.* Performance comparison on Qwen3-VL. CHAIR is evaluated under different max-token budgets; POPE reports averaged metrics.

| Method | Max Token = 64 | | Max Token = 128 | | Max Token = 512 | | POPE | | |
|---|---|---|---|---|---|---|---|---|---|
| | $C_S \downarrow$ | $C_I \downarrow$ | $C_S \downarrow$ | $C_I \downarrow$ | $C_S \downarrow$ | $C_I \downarrow$ | Acc ↑ | Rec ↑ | F1 ↑ |
| Regular | **8.00** | **3.35** | 17.00 | 5.79 | 45.00 | 9.06 | 89.05 | 85.00 | 88.62 |
| **Ours** | **8.00** | 3.77 | **14.00** | **5.43** | **43.00** | **7.93** | **89.48** | **85.69** | **89.10** |

We also evaluate REVIS on InternVL3 and InternVL3.5. These models utilize a distinct architecture compared to the Qwen and LLaVA series. As presented in Table 15, our method consistently mitigates hallucinations across both versions. For InternVL3, we observe that the discriminative performance on POPE remains comparable to the baseline (90.54% vs 90.24%). We attribute this plateau to the model reaching a capability ceiling on this specific discriminative task, where the room for marginal improvement is minimal. However, our method significantly enhances generative safety by reducing the $CHAIR_S$ score from 29.00% to 24.00%, proving that REVIS effectively corrects hallucinations during the generation process even when discriminative signals are saturated.

The performance on InternVL3.5 further validates this effectiveness, delivering comprehensive improvements across all metrics. Specifically, our method boosts the POPE Accuracy from 88.14% to 88.80% and dramatically lowers the $CHAIR_S$ score from 30.00% to 23.00%. This corresponds to a 23.3% reduction in the hallucination rate compared to the regular baseline. These consistent results confirm that the orthogonality between visual evidence and language priors is a fundamental property shared across diverse LVLM architectures, ensuring that our steering mechanism remains effective regardless of the underlying model backbone.

*Table 15.* Performance comparison on InternVL series models.

| Method | InternVL3 | | | | | InternVL3.5 | | | | |
| | POPE | | | CHAIR | | POPE | | | CHAIR | |
| | Acc ↑ | Rec ↑ | F1 ↑ | $C_S$ ↓ | $C_I$ ↓ | Acc ↑ | Rec ↑ | F1 ↑ | $C_S$ ↓ | $C_I$ ↓ |
|---|---|---|---|---|---|---|---|---|---|---|
| Regular | 90.54 | 90.98 | 90.65 | 29.00 | 8.44 | 88.14 | 92.51 | 88.78 | 30.00 | 9.09 |
| **Ours** | 90.24 | 92.18 | 90.53 | 24.00 | 7.86 | 88.80 | 89.53 | 88.97 | 23.00 | 5.73 |

## F.5. Empirical Validation of REVIS

We verify the effectiveness of REVIS across six model families: Qwen2.5-VL, Qwen3-VL, LLaVA-1.5, LLaVA-NeXT, InternVL3, and InternVL3.5. Across these architectures, the projected visual direction consistently reduces hallucination while preserving general utility, suggesting that the Gram-Schmidt projection is not tied to a particular backbone. We also find the projection to be well conditioned in practice. The main degenerate case would be a nearly parallel visual direction and language-prior direction, which could shrink the projected vector. This behavior does not arise in our experiments. This is expected by construction: the two directions are estimated from image-conditioned and text-only forward passes, so visual evidence induces a distinct source of variation and yields meaningful angular separation. When the two directions are already close to orthogonal, the projection is benign and retains most of the visual signal.

Furthermore, the sensitivity analysis in Figure 3 provides direct empirical evidence for this decoupling. The entangled raw vector becomes unstable under high-intensity intervention and can lead to degeneration, whereas the purified vector supports stronger intervention without model collapse. This contrast explains why orthogonal decoupling is a necessary component of REVIS rather than a cosmetic transformation of the steering direction.

## F.6. Robustness to Sequence Length on CHAIR

We investigate the robustness of REVIS against varying generation lengths by adjusting the maximum token limit (64, 128, and 512). This ablation is critical because hallucinations in LVLMs often exhibit a "snowball effect," where early errors accumulate in longer descriptions. As shown in Table 16, we observe a positive correlation between sequence length and hallucination rates across all methods.

**Short-Context Stability.** In the constrained setting of 64 tokens, REVIS demonstrates immediate effectiveness, achieving a $CHAIR_S$ score of 7.00%. This effectively halves the error rate of the Regular baseline (14.00%) and outperforms the second-best method, ONLY (8.00%), indicating strong initial grounding.

**Long-Context Robustness.** The advantage of REVIS becomes even more pronounced in long-form generation (512 tokens). While baselines like VCD and VTI suffer from severe degradation, rising to 34.00% and 35.00% respectively (worse than the Regular decoding at 31.00%), REVIS maintains a significantly lower error rate of 25.00%. This contrast indicates that while other methods fail to curb semantic drift over extended sequences, REVIS provides persistent and effective visual grounding throughout the entire decoding process.

*Table 16.* CHAIR results with different max token settings of Qwen2.5-VL.

| Method | Max Token = 64 | | Max Token = 128 | | Max Token = 512 | |
| | $CHAIR_S$ ↓ | $CHAIR_I$ ↓ | $CHAIR_S$ ↓ | $CHAIR_I$ ↓ | $CHAIR_S$ ↓ | $CHAIR_I$ ↓ |
|---|---|---|---|---|---|---|
| Regular* | 14.0 | 6.42 | 28.0 | 8.18 | 31.0 | 8.13 |
| VCD | 13.0 | 5.77 | 28.0 | 8.09 | 34.0 | 9.06 |
| M3ID | 11.0 | 3.94 | 23.0 | 6.65 | 31.0 | 8.88 |
| ONLY | 8.0 | 3.82 | 24.0 | 8.00 | 33.0 | 9.75 |
| AGLA | 10.0 | 3.93 | 23.0 | 7.22 | 32.0 | 9.15 |
| VTI | 9.0 | 3.91 | 24.0 | 6.35 | 35.0 | 7.49 |
| **Ours** | 7.0 | 3.40 | 20.0 | 7.39 | 25.0 | 8.23 |

# G. Detail Algorithm

We present the complete algorithmic procedure of REVIS in Algorithm 2. While the main text provides a condensed overview of the framework, this detailed version explicitly outlines the step-by-step mathematical operations for Orthogonal Visual Vector Construction, Calibration-based Layer Selection, and Inference-time Dynamic Steering.

---

**Algorithm 2** REVIS: Sparse Orthogonal Intervention

---

1: **Input:** LVLM $\mathcal{M}$, Extraction Set $\mathcal{D}_{\text{ext}}$, Calibration Set $\mathcal{D}_{\text{cal}}$, Scale $\alpha$, Percentile $k$
2: **Output:** Generated Token Sequence $\mathbf{y}$

3: *// Phase 1: Orthogonal Visual Vector Construction (§4.1)*
4: **for** $\ell = 1$ **to** $L$ **do**
5:     {Extract Raw Visual & Language Prior Vectors}
6:     $\mathbf{v}_{\text{raw}}^{(\ell)} \leftarrow \mathbb{E}_{\mathcal{D}_{\text{ext}}}[\mathbf{h}_{\text{gt}}^{(\ell)} - \mathbf{h}_{\emptyset\_\text{gt}}^{(\ell)}]; \quad \mathbf{v}_{\text{prior}}^{(\ell)} \leftarrow \mathbb{E}_{\mathcal{D}_{\text{ext}}}[\mathbf{h}_{\emptyset\_\text{hall}}^{(\ell)} - \mathbf{h}_{\emptyset\_\text{unk}}^{(\ell)}]$
7:     {Gram-Schmidt Orthogonalization}
8:     $\mathbf{v}_{\text{vis}}^{\perp(\ell)} \leftarrow \mathbf{v}_{\text{raw}}^{(\ell)} - \frac{\mathbf{v}_{\text{raw}}^{(\ell)} \cdot \mathbf{v}_{\text{prior}}^{(\ell)}}{\|\mathbf{v}_{\text{prior}}^{(\ell)}\|^2} \mathbf{v}_{\text{prior}}^{(\ell)}$
9: **end for**

10: *// Phase 2: Layer Selection (§4.2)*
11: Extract hidden state sets $\mathcal{H}_{\text{fact}}^{(\ell)}$ and $\mathcal{H}_{\text{hall}}^{(\ell)}$ from $\mathcal{D}_{\text{cal}}$ via POPE protocols
12: **for** $\ell = L$ **down to** $1$ **do**
13:     $R(\mathbf{h}) \leftarrow -\text{CosSim}(\mathbf{h}, \mathbf{v}_{\text{vis}}^{\perp(\ell)})$
14:     $\Delta_\ell \leftarrow R(\mathcal{H}_{\text{hall}}^{(\ell)}) - R(\mathcal{H}_{\text{fact}}^{(\ell)})$
15:     {Check Geometric Consistency}
16:     **if** $\Delta_\ell > 0$ **then**
17:         $L^* \leftarrow \ell$
18:         **break** {Select the deepest valid layer}
19:     **end if**
20: **end for**
21: $L^* \leftarrow \arg\max_\ell \Delta_\ell$
22: $\tau \leftarrow \text{Percentile}(\{R(\mathbf{h}) \mid \mathbf{h} \in \mathcal{H}_{\text{fact}}^{(L^*)}\}, k)$ {Set Safety Threshold}

23: *// Phase 3: Dynamic Inference (§4.3)*
24: **while** not EOS **do**
25:     $\mathbf{h}_t \leftarrow \mathcal{M}.\text{encode}(x_{<t})$ at layer $L^*$
26:     $R_t \leftarrow -\text{CosSim}(\mathbf{h}_t, \mathbf{v}_{\text{vis}}^{\perp(L^*)})$
27:     $\lambda_t \leftarrow \alpha \cdot \mathbb{1}(R_t - \tau)$ {Risk-Aware Gating}
28:     $\tilde{\mathbf{h}}_t \leftarrow \mathbf{h}_t + \lambda_t \mathbf{v}_{\text{vis}}^{\perp(L^*)}$ {Visual Re-activation}
29:     $y_t \sim \text{Softmax}(\mathcal{M}.\text{head}(\tilde{\mathbf{h}}_t))$
30:     Append $y_t$ to output $\mathbf{y}$
31: **end while**

---

