# OpenReview forum: "REVIS: Sparse Latent Steering to Mitigate Object Hallucination in Large Vision-Language Models"
_ICML.cc/2026/Conference — ICML 2026 regular_

### Official Review · Reviewer_X8C2 · 2026-03-10

**Soundness:** 3
**Presentation:** 3
**Significance:** 2
**Originality:** 2
**Overall Recommendation:** 4
**Confidence:** 3

**Summary:**

The paper proposes an inference-time algorithm to reduce object hallucinations in large vision-language models. They first conduct a preliminary analysis of the cause of hallucination, then propose to isolate visual semantics from the model’s language priors. Based on the extracted visual vectors, they select one layer to conduct a proposed steering to modify the hidden states during inference. Experiments show improvements on hallucination and general benchmarks.

**Compliance With Llm Reviewing Policy:**

Affirmed.

**Final Justification:**

Most of my concerns have been addressed, thus I am increasing my score. It would be good to include the size of qwen3-vl in the added experiments and add more benchmarks.

**Key Questions For Authors:**

1. Have you tried more models such as Qwen3VL?
2. Have you tried more benchmarks such as MMMU-pro?
3. In Table 7, can you provide results for different models?

**Limitations:**

yes

**Strengths And Weaknesses:**

Strengths:
1. The idea is intuitive and the proposed method is techinically sound. The idea of decoupling visual inputs and language priors is intuitive and the paper proposes a valid way to solve this. The preliminary analysis well motivates their method.
2. The experimental results on their benchmarks are good, showing improvements on both general and hallucination benchmarks.
3. The presentation is clearly written and well structured. The figures are informative and can well illustrate their ideas.


Weaknesses:
1. More general benchmarks should be tested. Since the method is an inference-only method, there are lots of existing benchmarks that should be tested such as MMMU-pro.
2. More baselines should be tested. LLaVA-1.5 and LLaVA-NeXT are relatively old models and state-of-the-art models such as Qwen3-VL should be used.
3. It seems that selecting the optimal layer is quite important and the optimal layer is model-specific. Thus, it is unclear whether the method is robust or not.

---

> ### Author Rebuttal · Authors · 2026-03-29
>
> We thank Reviewer X8C2 for the constructive feedback. We address each concern below.
>
> ### **W2 & Q1: More state-of-the-art models should be tested (e.g., Qwen3-VL)**
>
> We respectfully clarify that our evaluation already covers a broad range of recent architectures beyond LLaVA-1.5 and LLaVA-NeXT:
> - **Qwen2.5-VL** (Table 3 and Table 4 in the main paper)
> - **InternVL3 and InternVL3.5** (Table 12 in the appendix)
> - **Qwen2.5VL-32B** (newly added, see Reviewer yEqX W1)
>
> Notably, **InternVL3.5's LLM backbone is Qwen3**, providing direct coverage of the latest language model generation. Additionally, we have added experiments on **Qwen3-VL** in the revised manuscript to directly address this concern:
>
> **CHAIR & POPE results on Qwen3-VL:**
>
> | Method | CHAIRs (64) % | CHAIRi (64) % | CHAIRs (128) % | CHAIRi (128) % | CHAIRs (512) % | CHAIRi (512) % | POPE Acc % | POPE Recall % | POPE F1 % |
> |--------|:-----------:|:-----------:|:------------:|:------------:|:------------:|:------------:|:--------:|:-----------:|:-------:|
> | Regular | **8** | **3.35** | 17 | 5.79 | 45 | 9.06 | 89.05 | 85.00 | 88.62 |
> | REVIS  | **8** | 3.77 | **14** | **5.43** | **43** | **7.93** | **89.48** | **85.69** | **89.10** |
>
> REVIS reduces hallucination on Qwen3-VL at longer generation lengths (128 and 512 tokens), where hallucination accumulation is more pronounced and practically relevant. At the shorter 64-token setting, the base model already exhibits very low hallucination rates (CHAIRi = 3.35), leaving limited room for improvement. Meanwhile, REVIS consistently improves all POPE metrics (Accuracy, Recall, F1), confirming its effectiveness on this latest-generation architecture.
>
> ### **W1 & Q2: More general benchmarks should be tested (e.g., MMMU-Pro)**
>
> We have now added **MMMU-Pro** results across four models:
>
> | Method | Qwen2.5VL-7B (%)|||| Qwen2.5VL-32B (%)|||| Qwen3-VL (%)|||| LLaVA-NEXT (%)||||
> |--------|:---:|:---:|:---:|:---:|:---:|:---:|:---:|:---:|:---:|:---:|:---:|:---:|:---:|:---:|:---:|:---:|
> | | 4-opt  | 10-opt | vision | avg | 4-opt | 10-opt | vision | avg | 4-opt | 10-opt | vision | avg | 4-opt | 10-opt | vision | avg |
> | Regular | 41 | 33.5 | 30 | 34.8 | 51 | 39 | 36.5 | 42.2 | 46 | 37 | 31 | 38 | 41.5 | 19.5 | 13.5 | 24.8 |
> | REVIS | **42** | 31.5 | **33.5** | **35.7** | **52** | **41** | **37.5** | **43.5** | **48** | **40** | 29.5 | **39.2** | 41.5 | 19.5 | 13.5 | 24.8 |
>
> On Qwen2.5VL-7B, Qwen2.5VL-32B, and Qwen3-VL, REVIS consistently improves MMMU-Pro average scores, demonstrating that our intervention enhances visual grounding without hurting general multi-modal reasoning. On LLaVA-NEXT, performance is preserved (no degradation), which is expected since REVIS's risk-aware mechanism avoids unnecessary intervention when the model is already confident.
>
> ### **W3 & Q3: Optimal layer selection is model-specific; robustness unclear / Table 7 results for different models**
>
> The optimal layer is indeed model-specific; however, we provide results for different models to demonstrate robustness. In addition to Table 7 in the main paper, detailed per-model results are available in the appendix: **LLaVA-NeXT** (Table 10) and **LLaVA-1.5** (Table 11). These tables show that REVIS performs consistently across architectures regardless of the specific optimal layer chosen.
>
> Furthermore, we observe a clear pattern: **the optimal layer consistently lies near the final layers** across all tested architectures (last layer for Qwen2.5VL-7B, second-to-last for LLaVA-1.5/NeXT, third-to-last for Qwen2.5VL-32B). This consistency is well-explained by our mechanistic analysis, language priors increasingly dominate in deeper layers. The calibration procedure is fully automated, lightweight (single forward pass over a small calibration set), and a one-time cost per model.
>
> We thank the reviewer again for the valuable feedback. We hope our response addresses your concerns, potentially leading to a higher score.

---

> > ### Author Rebuttal · Reviewer_X8C2 · 2026-04-04
> >
> > Most of my concerns have been addressed, thus I am increasing my score. It would be good to include the size of qwen3-vl in the added experiments and add more benchmarks.

---

> > > ### Author Response · Authors · 2026-04-04
> > >
> > > We sincerely thank you for the positive feedback and for increasing the score. We would like to clarify that Qwen3-VL-8B was used for the additional experiments. We apologize for the oversight in our initial rebuttal.
> > >
> > > Thank you again for your constructive guidance in improving our work.

---

### Official Review · Reviewer_H8jh · 2026-03-11

**Soundness:** 3
**Presentation:** 3
**Significance:** 3
**Originality:** 3
**Overall Recommendation:** 4
**Confidence:** 4

**Summary:**

This paper proposes REVIS, a training-free sparse latent steering framework for mitigating object hallucination in large vision-language models. The paper is motivated by the hypothesis that hallucination is caused by feature entanglement in the latent space: visual information is suppressed by dominant language priors, so naïvely steering the model can amplify both grounded visual evidence and hallucination-inducing priors at the same time. To address this, REVIS uses orthogonal projection to extract a pure visual information vector, performs sparse intervention at an automatically selected layer where factual and hallucinatory states are most separable, and activates the intervention only when a latent hallucination risk score is high. Experiments on standard benchmarks suggest that the method improves both hallucination mitigation and inference efficiency relative to prior baselines.

**Compliance With Llm Reviewing Policy:**

Affirmed.

**Key Questions For Authors:**

Why the improvements look very marginal? Does that mean the hallucination problem have been somehow solved?

**Limitations:**

yes

**Strengths And Weaknesses:**

Strengthens:
    1. The paper is well motivated and tackles an important problem. Object hallucination in LVLMs is a central and practically important issue, and the paper goes beyond surface-level output correction by framing the problem in terms of latent visual-textual conflict.
    2. The proposed method is conceptually clear and technically coherent. The design of REVIS is easy to follow: it first identifies the entanglement problem, then isolates a pure visual direction via orthogonalization, selects an effective intervention layer through calibration, and finally applies risk-aware sparse steering at inference time.
    3. The method is attractive from an efficiency standpoint. Compared with prior inference-time approaches that require multiple forward passes or auxiliary procedures, REVIS is positioned as a lightweight intervention method with minimal additional cost.

Weaknesses:
    1. Although the paper introduces orthogonal projection and sparse latent intervention, the overall paradigm still falls within the broader line of activation steering / inference-time correction methods. Compared with prior approaches that already mitigate hallucinations through decoding-time intervention, logit adjustment, or latent manipulation, the present work seems more like a refined variant than a fundamentally new direction.
    2. The paper positions REVIS as a lightweight alternative to methods such as VCD, M3ID, ONLY, and AGLA, with a clear efficiency benefit, especially in inference latency. However, from a reviewer perspective, the key question is whether the performance gains over these strong baselines are consistently large enough to establish clear superiority, rather than mainly offering a better efficiency–effectiveness trade-off.

---

> ### Author Rebuttal · Authors · 2026-03-29
>
> We thank Reviewer H8jh for the positive assessment and thoughtful questions.
>
> ### **W1: The overall paradigm falls within activation steering / inference-time correction; more of a refined variant than a fundamentally new direction**
>
> We respectfully note that while REVIS belongs to the inference-time intervention family, our contribution addresses a previously unidentified failure mode that fundamentally limits existing methods. We summarize the key distinctions from prior work:
>
> **(1) Identifying feature entanglement as the root cause.** Our analysis (Section 3) reveals that while factual and hallucinatory states are linearly separable in deep layers, the raw difference vector remains entangled with dominant language priors. Naively amplifying it causes model collapse by boosting both visual evidence and hallucination-inducing priors. Our feature entanglement analysis (Section 3.2) confirms that only purified latent directions enable stable, high-intensity intervention. This entanglement failure mode is unrecognized by prior work.
>
> **(2) Orthogonal decoupling as a principled solution.** Prior methods (VCD, M3ID, ONLY, AGLA) operate at the output level (logit adjustment, contrastive decoding) or steer with entangled representations. REVIS instead intervenes on a purified visual vector obtained via orthogonal projection, enabling high-intensity steering without degradation (Figure 3).
>
> **(3) Sparse, risk-aware intervention.** REVIS intervenes at a single automatically selected layer and activates only when the latent hallucination risk exceeds a threshold, rather than applying dense or unconditional corrections.
>
> Together, these constitute a new diagnosis, a new intervention target, and a new activation strategy, advancing beyond refinement of existing approaches.
>
> ### **W2: Whether performance gains are consistently large enough to establish clear superiority**
>
> We emphasize the **consistency** of REVIS's improvements across a comprehensive evaluation. Our method has been validated on **six model families** (Qwen2.5-VL, Qwen3-VL, LLaVA-1.5, LLaVA-NeXT, InternVL3, InternVL3.5), across **multiple benchmarks** (CHAIR, POPE, MM-Vet, MME, MMMU-Pro), and at **multiple model scales**. During the limited time of the rebuttal period, we have added new benchmark results on Qwen2.5VL-32B as shown below:
>
> | Method | CHAIRs (64) % | CHAIRi (64) % | CHAIRs (128) % | CHAIRi (128) % | CHAIRs (512) % | CHAIRi (512) % | MM-Vet | MMMU-Pro (4-opt) % | MMMU-Pro (10-opt) % | MMMU-Pro (vision) % |
> |--------|:-----------:|:-----------:|:------------:|:------------:|:------------:|:------------:|:------:|:----------------:|:-----------------:|:-----------------:|
> | Regular | 9 | 5.26 | 22 | 6.72 | 52 | 11.05 | 74.31 | 51 | 39 | 36.5 |
> | REVIS  | **8** | **4.91** | **18** | **5.85** | **46** | **8.99** | 74.13 | **52** | **41** | **37.5** |
>
> Across all these settings, REVIS delivers consistent improvements on hallucination metrics while preserving or enhancing general reasoning capabilities. This breadth and consistency of evaluation, spanning diverse architectures, model scales, and evaluation protocols, demonstrates that REVIS's gains are not benchmark- or model-specific, but reflect a reliable and generalizable advance.
>
> ### **Q: Why do the improvements look very marginal? Does that mean the hallucination problem has been somehow solved?**
>
> The hallucination problem is far from solved. The primary goal of our work is to investigate whether **language prior** is a main causal factor behind hallucination and to provide both an explanation and a targeted solution. There remain many unexplored dimensions of the hallucination problem. Moreover, stronger base models are themselves entering a phase of diminishing marginal returns on hallucination mitigation, and there is no established consensus or unified standard on what degree of improvement constitutes "solving" hallucination. We view REVIS as advancing the mechanistic understanding of *why* hallucination occurs and providing a principled, lightweight intervention that complements ongoing model improvements.
>
> We thank the reviewer again for the valuable feedback. We hope our response addresses your concerns, potentially leading to a higher score.

---

### Official Review · Reviewer_yEqX · 2026-03-13

**Soundness:** 3
**Presentation:** 3
**Significance:** 3
**Originality:** 3
**Overall Recommendation:** 4
**Confidence:** 3

**Summary:**

This paper tackles object hallucination in Large Vision-Language Models, identifying the root cause as the entanglement of visual features with dominant language priors in deep network layers. The authors propose REVIS, a training-free framework that employs sparse latent steering to re-activate suppressed visual information while preserving general reasoning capabilities. REVIS extracts a purified visual vector via orthogonal projection and identifies a single optimal intervention layer through calibration-based geometric search. At inference time, a dynamic risk-aware mechanism activates steering only when potential hallucination is detected, incurring negligible computational overhead. Extensive experiments demonstrate that REVIS reduces hallucination rates by approximately 19% compared to state-of-the-art baselines across diverse architectures including Qwen2.5-VL, LLaVA, and InternVL.

**Compliance With Llm Reviewing Policy:**

Affirmed.

**Key Questions For Authors:**

1. When raw visual and language prior vectors are nearly parallel or orthogonal, does Gram-Schmidt projection fail? What mathematical conditions guarantee decoupling effectiveness or upper bounds of feature entanglement?

2. Is optimal layer $L^*$ strongly architecture-dependent? Must full calibration be repeated for different vision encoders or LLM backbones, or can it transfer across models?

3. $\tau$ is determined by percentile of factual distributions from calibration data. When input distribution shifts (e.g., natural to medical images), does $\tau$ remain effective at distinguishing hallucination from factual states?

**Limitations:**

No. While Section 7 mentions model scale limitations, critical limitations remain underdiscussed.

**Strengths And Weaknesses:**

Strengths：
1. The orthogonal projection mechanism offers a clear geometric interpretation, explicitly separating visual information from language priors via Gram-Schmidt process to overcome feature entanglement-induced model collapse, validated across Qwen2.5-VL, LLaVA, and InternVL architectures.
2. The sparse intervention strategy significantly reduces computational overhead, requiring only single-layer dynamic activation to outperform dense interventions, with inference latency comparable to standard decoding and substantially lower than contrastive decoding methods requiring parallel forward passes.
3. The dynamic risk-gating mechanism enables precise intervention, activating corrections only when potential hallucination is detected, preserving general reasoning capabilities (improving MM-Vet scores) while reducing hallucination rates by approximately 19%.

Weaknesses
1. Limited evaluation to 7B/8B models without validation on larger scales, raising scalability concerns.
2. Optimal layer selection relies on calibration datasets; whether recalibration is required per model or task in practical deployment remains unclear.
3. Insufficient theoretical guarantees for orthogonal projection effectiveness, relying primarily on empirical observations without mathematical bounds or convergence analysis for feature decoupling.

---

> ### Author Rebuttal · Authors · 2026-03-29
>
> We thank Reviewer yEqX for the thorough evaluation and constructive questions.
>
> ### **W1 & Scalability: Limited evaluation to 7B/8B models without validation on larger scales**
>
> We have now extended our evaluation to **Qwen2.5VL-32B** across three benchmarks (CHAIR, MM-Vet, MMMU-Pro), confirming the scalability of REVIS:
>
> | Method | CHAIRs (64) % | CHAIRi (64) % | CHAIRs (128) % | CHAIRi (128) % | CHAIRs (512) % | CHAIRi (512) % | MM-Vet | MMMU-Pro (4-opt) % | MMMU-Pro (10-opt) % | MMMU-Pro (vision) % |
> |--------|:-----------:|:-----------:|:------------:|:------------:|:------------:|:------------:|:------:|:----------------:|:-----------------:|:-----------------:|
> | Regular | 9 | 5.26 | 22 | 6.72 | 52 | 11.05 | 74.31 | 51 | 39 | 36.5 |
> | REVIS  | **8** | **4.91** | **18** | **5.85** | **46** | **8.99** | 74.13 | **52** | **41** | **37.5** |
>
> REVIS consistently reduces hallucination (CHAIR) at the 32B scale while preserving or improving general reasoning (MM-Vet, MMMU-Pro). This demonstrates that our method scales effectively beyond 7B/8B models.
>
> ### **W3 & Q1: Theoretical guarantees for orthogonal projection / Does Gram-Schmidt fail when vectors are nearly parallel or orthogonal?**
>
> We have verified the effectiveness of Gram-Schmidt projection across **six** model families (Qwen2.5-VL, Qwen3-VL, LLaVA-1.5, LLaVA-NeXT, InternVL3, and InternVL3.5) and confirm that the degenerate case (nearly parallel or orthogonal) does not arise in practice. This is expected by construction: the two vectors are derived from *image-conditioned* vs. *text-only* forward passes, so the visual input naturally places them in distinct subspaces with meaningful angular separation.
>
> Conceptually, the effectiveness of orthogonal decoupling is supported by the principles of **null-space projection** [1] and **task arithmetic** [2,3], which have been extensively validated in the model editing and model merging literature. These works demonstrate that projecting out interfering components reliably isolates desired behavioral directions. In our case, the language prior is the interfering component, and our projection removes it to isolate the pure visual direction.
>
> Furthermore, the ablation comparing models **with and without** Gram-Schmidt projection (Figure 3) provides direct causal evidence: the purified vector enables high-intensity intervention without model collapse, while the entangled vector leads to degeneration.
>
> We acknowledge that, as with the broader latent steering paradigm, our approach remains empirical rather than providing formal mathematical bounds. We will add a discussion of these points in the revised manuscript.
>
> ### **W2 & Q2: Architecture-dependent optimal layer / Whether calibration can transfer across models**
>
> Due to differences in base model pretraining, post-training, and architecture, the model families we evaluate, LLaVA, Qwen2.5-VL, Qwen3-VL, and InternVL, employ different vision encoders and LLM backbones. While our method generalizes across all these architectures, the hidden states extracted from different models are inherently model-specific and cannot be directly transferred.
>
> However, we argue that the concern about transferability aligns with our goal of minimizing overhead,the layer selection is fully automated and imposes **no additional burden** on users. Specifically:
> - The layer search is lightweight: for Qwen2.5VL-7B, the optimal layer is the last layer; for LLaVA-1.5 and LLaVA-NeXT, it is the second-to-last layer; for the larger Qwen2.5VL-32B, it is layer 62 (the third-to-last layer).
> - Computing the steering vector requires only a small set of calibration samples and a single forward pass.
> - The entire calibration is a **one-time cost** per model, and the resulting vectors can be reused across all subsequent inference tasks.
>
> ### **Q3: Does τ remain effective under distribution shift?**
>
> We use a **unified percentile** k = 0.8 across all models and experiments (Section 4.2): τ = Percentile(S_fact, k=0.8). The actual τ value is model-specific (since different models produce different latent risk scores), but k is fixed and requires no per-task tuning.
>
> Our calibration data is sampled from **COCO-train**, while evaluation benchmarks include substantially different distributions (e.g., **MM-Vet** contains OCR, mathematical reasoning, and diverse image types entirely absent from COCO-train). The strong performance across all these out-of-distribution benchmarks with the same k=0.8 demonstrates that τ generalizes well under distribution shift, as it captures a fundamental geometric property of the latent space (the boundary between factual and hallucinatory states) rather than surface-level distributional characteristics.
>
> We hope our response addresses your concerns.
>
> Reference：
>
> [1] AlphaEdit: Null-Space Constrained Knowledge Editing for Language Models.
>
> [2] Editing Models with Task Arithmetic.
>
> [3] Task Arithmetic in the Tangent Space: Improved Editing of Pre-Trained Models.

---

> > ### Author Rebuttal · Reviewer_yEqX · 2026-04-07
> >
> > Thank the authors for their great efforts in addressing all my concerns. Most of my concerns and questions have been resolved.

---

> > > ### Author Response · Authors · 2026-04-07
> > >
> > > Thank you for your kind reply and for confirming that your concerns have been fully resolved. We truly appreciate your constructive feedback, which has helped us significantly improve the paper.
> > >
> > > As our rebuttal has addressed your questions, we would be very grateful if you might consider raising your rating.
> > >
> > > Thank you again for your time and valuable guidance!

---

### Decision · Program_Chairs · 2026-04-30

**Decision:**

Accept (regular)

**Comment:**

This paper received consistently positive comments (WA/WA/WA). All reviewers agreed that object hallucination in LVLMs is an important issue, and REVIS is well-motivated and attractive from efficiency aspects.   The rebuttal effectively addressed most of the concerns by including larger-scale models and broader benchmark results, and clarified the pattern regarding model robustness. Performance is well-delivered, reducing object hallucination rates by 19% compared to without compromising general reasoning capabilities.